# Excitonic Bloch equations from first principles

Gianluca Stefanucci[1,2], and Enrico Perfetto[1,2]

**1** Dipartimento di Fisica, Università di Roma Tor Vergata, Via della Ricerca Scientifica 1, 00133 Rome, Italy

**2** INFN, Sezione di Roma Tor Vergata, Via della Ricerca Scientifica 1, 00133 Rome, Italy

⋆ gianluca.stefanucci@roma2.infn

July 25, 2024

## Abstract

The ultrafast conversion of coherent excitons into incoherent excitons, as well as the subsequent exciton diffusion and thermalization, are central topics in current scientific research due to their relevance in optoelectronics, photovoltaics and photocatalysis. Current approaches to the exciton dynamics rely on *model* Hamiltonians that depend on already screened electron-electron and electron-phonon couplings. In this work, we subject the state-of-the-art methods to scrutiny using the *ab initio* Hamiltonian for electrons and phonons. We offer a rigorous and intuitive proof demonstrating that the exciton dynamics governed by model Hamiltonians is affected by an overscreening of the electron-phonon interaction. The introduction of an auxiliary exciton species, termed the irreducible exciton, enables us to formulate a theory free from overscreening and derive the excitonic Bloch equations. These equations describe the time-evolution of coherent, irreducible, and incoherent excitons during and after the optical excitation. They are applicable beyond the linear regime, and predict that the total number of excitons is preserved when the external fields are switched off.

# 1  Introduction

Transition-metal dichalcogenides, layered heterostructures, and other two-dimensional materials hold great promise for next-generation optoelectronic devices due to their rich excitonic landscape [1–5]. The weak dielectric screening favors the existence (and coexistence) of well defined optically bright zero-momentum singlet excitons [6–8] as well as dark intervalley and triplet excitons [9–12]. Understanding the fundamental laws governing the formation and scattering of excitons is crucial for guiding researchers and accelerating progress. The momentum-resolved exciton dynamics can be monitored using time-resolved photoemission techniques [13–26]. Laser pulses with subgap frequencies excite particle-hole pairs and transfer the light's coherence to them. During and shortly after the optical excitation, the nonequilibrium state features coherent (or virtual) particle-hole pairs bound by the Coulomb attraction, i.e., bright coherent excitons. For not too large excitation densities, this superfluid phase [27–31] is stable only at clamped nuclei [32]. The charge imbalance created by the laser pulse sets the nuclear lattice in motion, and the macroscopic number of lattice modes (or phonons) is responsible for destroying exciton coherence [33–38] and diffusing excitons [39–41]. Bright coherent excitons are then converted into bright and dark incoherent excitons, whose energy and momentum can be inferred from the measured time-resolved and angle-resolved photocurrent.

The complex dynamics of coherent and incoherent excitons coupled to phonons has become a central focus in current scientific research. A straightforward strategy to deal with the problem consists of using model bosonic Hamiltonians for excitons and phonons. Built on the pioneering works by Toyozawa and Hopfield [42–44], this approach dates back to the early sixties [45–47]. Nonetheless, it faces important conceptual issues that remain to be addressed [48–50]. The major issue pertains with the fact that model Hamiltonians are written in terms of screened electron-electron (*e-e*) and electron-phonon (*e-ph*) interactions. This leads to double counting and overscreening already at the leading order in perturbation theory. Techniques based on the cluster expansion [51–55] suffer from the same conceptual drawbacks, as the underlying Hamiltonian is affected by the same problems, i.e., a screened *e-e* and *e-ph* coupling. The equivalence between the two approaches becomes evident when expanding the *e-ph* interaction in terms of an interaction between (multiple) particle-holes and phonons [56, 57], which, to the lowest order, is equivalent to a bosonization.

Despite the conceptual issues, the equations of motion for the coupled dynamics of coherent and incoherent excitons [52–54, 58, 59] have a great appeal, and have been extensively used in the literature. The fundamental question we address in this work is: Can these equations be derived from the ab initio Hamiltonian of electrons and phonons [60]? In this study, we provide a conclusive negative answer, pinpointing the issue as rooted in the overscreening of the *e-ph* interaction, and derive the correct result. Our derivation is based on Green's function many-body theory, where screening, quasi-particle renormalization, and phonon frequencies emerge naturally from the diagrammatic treatment of the ab initio Hamiltonian of electrons and phonons.

The paper is organized as follows. In Section 2 we introduce the connection between excitons and two-particle Green's function, and briefly go through the ab initio Hamiltonian for electrons and phonons. We also present key properties of the Bethe-Salpeter equation for systems with *e-e* and *e-ph* interactions. In Sections 3 and 4 we discuss the minimal approximation to the self-energy and exchange-correlation kernel leading to a coupled dynamics of coherent and incoherent excitons. The inclusion of the inelastic exciton-phonnon scattering is elaborated in Section 5. The final outcome, summarized in Section 6, is the excitonic Bloch equations – a comprehensive set of coupled equations for coherent, irreducible and incoherent excitons. Finally, in Section 7 we provide a summary of our main findings and outline future avenues for research and development.

## 2  Preliminaries

### 2.1  Excitons and two-particle Green's function

We consider a semiconductor with quasiparticle energies $\epsilon_{c\mathbf{k}}$ for conduction electrons in band $c$ with momentum $\mathbf{k}$, and $\epsilon_{v\mathbf{k}}$ for valence electrons in band $v$ with momentum $\mathbf{k}$. In the quasiparticle basis the Coulomb amplitude for the scattering process $(\mu\mathbf{k}+\mathbf{Q}, v'\mathbf{k}') \rightarrow (\mu'\mathbf{k}'+\mathbf{Q}, v\mathbf{k})$ reads

$$v_{\mu\mathbf{k}+\mathbf{Q}\,v'\mathbf{k}'\,v\mathbf{k}\,\mu'\mathbf{k}'+\mathbf{Q}} = \langle \mu\mathbf{k}+\mathbf{Q}\,v'\mathbf{k}'|\frac{1}{|\hat{\mathbf{r}}-\hat{\mathbf{r}}'|}|v\mathbf{k}\,\mu'\mathbf{k}'+\mathbf{Q}\rangle. \tag{1}$$

We normalize the quasiparticle wavefunctions to unity so that the Coulomb amplitude scales like $1/\mathcal{N}_\mathbf{k}$, $\mathcal{N}_\mathbf{k}$ being the number of $\mathbf{k}$-points in the first Brillouin zone. The exciton is a bound electron-hole pair of the crystal with clamped nuclei, whose energy and eigenfunction satisfy the eigenvalue equation

$$E^\mathbf{Q}_{cv\mathbf{k}}A^{\lambda\mathbf{Q}}_{cv\mathbf{k}} - \sum_{c'v'\mathbf{k}'} K^{\mathrm{HSEX},\mathbf{Q}}_{cv\mathbf{k},c'v'\mathbf{k}'}A^{\lambda\mathbf{Q}}_{c'v'\mathbf{k}'} = E_{\lambda\mathbf{Q}}A^{\lambda\mathbf{Q}}_{cv\mathbf{k}}, \tag{2}$$

where $E^\mathbf{Q}_{cv\mathbf{k}} = \epsilon_{c\mathbf{k}+\mathbf{Q}} - \epsilon_{v\mathbf{k}}$ and

$$K^{\mathrm{HSEX},\mathbf{Q}}_{cv\mathbf{k},c'v'\mathbf{k}'} = W_{c\mathbf{k}+\mathbf{Q}\,v'\mathbf{k}'\,v\mathbf{k}\,c'\mathbf{k}'+\mathbf{Q}} - v_{c\mathbf{k}+\mathbf{Q}\,v'\mathbf{k}'\,c'\mathbf{k}'+\mathbf{Q}\,v\mathbf{k}} = W^\mathbf{Q}_{cv\mathbf{k},c'v'\mathbf{k}'} - v_{c\mathbf{k}+\mathbf{Q}\,v'\mathbf{k}'\,c'\mathbf{k}'+\mathbf{Q}\,v\mathbf{k}} \tag{3}$$

is the irreducible Hartree plus screened exchange (HSEX) kernel, $W$ being the statically screened interaction [61,62]. At any fixed $\mathbf{Q}$ the exciton wavefunctions are chosen orthonormal, i.e., $\sum_{cv\mathbf{k}}A^{\lambda\mathbf{Q}*}_{cv\mathbf{k}}A^{\lambda'\mathbf{Q}}_{cv\mathbf{k}} = \delta_{\lambda\lambda'}$. Henceforth we use the word excitons for all solutions (bound and unbound) of Eq. (2). We define the exciton creation operator

$$\hat{X}^\dagger_{\lambda\mathbf{Q}} = \sum_{cv\mathbf{k}}A^{\lambda\mathbf{Q}}_{cv\mathbf{k}}\hat{d}^\dagger_{c\mathbf{k}+\mathbf{Q}}\hat{d}_{v\mathbf{k}}, \tag{4}$$

where the operators $\hat{d}_{\mu\mathbf{k}}$ annihilate an electron of momentum $\mathbf{k}$ in band $\mu$, and satisfy the anticommutation relations $\{\hat{d}_{\mu\mathbf{k}}, \hat{d}^\dagger_{\mu'\mathbf{k}'}\} = \delta_{\mu\mu'}\delta_{\mathbf{k},\mathbf{k}'}$. The number operator for excitons of type $\lambda$ and momentum $\mathbf{Q}$ is then

$$\hat{N}_{\lambda\mathbf{Q}} = \hat{X}^\dagger_{\lambda\mathbf{Q}}\hat{X}_{\lambda\mathbf{Q}}. \tag{5}$$

Looking at Eq. (2) it is tempting to construct an effective boson Hamiltonian for excitons, i.e., $\hat{H}^\mathrm{x} = \sum_{\lambda\mathbf{Q}}E_{\lambda\mathbf{Q}}\hat{X}^\dagger_{\lambda\mathbf{Q}}\hat{X}_{\lambda\mathbf{Q}}$. However, the eigenvalues $E_{\lambda\mathbf{Q}}$ depend on the irreducible HSEX kernel which, in turn, is the difference between screened and bare scattering amplitudes, see Eq. (3).

As we see below, avoiding the double counting of the (already incorporated) screening in a perturbative theory of bosonized excitons and phonons is a complex and delicate task. We therefore stick to the Green's function many-body theory, from which Eq. (2) naturally emerges when solving the Bethe-Salpeter equation in the HSEX approximation [62, 63].

The time-dependent average of $\hat{N}_{\lambda\mathbf{Q}}$ can be calculated from the 2-particle Green's function [64]

$$G_2(iz_i, jz_j; mz_m, nz_n) \equiv -\langle \mathcal{T}\{\hat{d}_i(z_i)\hat{d}_j(z_j)\hat{d}_n^\dagger(z_n)\hat{d}_m^\dagger(z_m)\}\rangle, \tag{6}$$

where the $z$'s are times on the Keldysh contour $C = C_- \cup C_+ = (0, \infty) \cup (\infty, 0)$ and $\mathcal{T}$ is the contour ordering operator. The indices $i, j, m, n$ carried by the electronic operators in Eq. (6) specify band and momentum. Consider the two-time propagator

$$\begin{aligned}
N_{c v \mathbf{k}, c' v' \mathbf{k}'}^{\mathbf{Q}}(z, z') &\equiv -G_2(c\mathbf{k}+\mathbf{Q}z, v'\mathbf{k}'z'; v\mathbf{k}z^+, c'\mathbf{k}'+\mathbf{Q}z^+) \\
&= \langle \mathcal{T}\{\hat{d}_{v\mathbf{k}}^\dagger(z)\hat{d}_{c\mathbf{k}+\mathbf{Q}}(z)\hat{d}_{c'\mathbf{k}'+\mathbf{Q}}^\dagger(z')\hat{d}_{v'\mathbf{k}'}(z')\}\rangle.
\end{aligned} \tag{7}$$

Rotating this quantity in the excitonic basis we obtain the *exciton Green's function*

$$N_{\lambda\mathbf{Q}}(z, z') = \sum_{\substack{cv\mathbf{k} \\ c'v'\mathbf{k}'}} A_{cv\mathbf{k}}^{\lambda\mathbf{Q}*} N_{cv\mathbf{k},c'v'\mathbf{k}'}^{\mathbf{Q}}(z, z') A_{c'v'\mathbf{k}'}^{\lambda\mathbf{Q}} = \langle \mathcal{T}\{\hat{X}_{\lambda\mathbf{Q}}(z)\hat{X}_{\lambda'\mathbf{Q}}^\dagger(z')\}\rangle. \tag{8}$$

For any real time $t$ we use the notation $z = t_\pm$ if $z \in C_\pm$. The average number of excitons is simply given by

$$N_{\lambda\mathbf{Q}}(t) = N_{\lambda\mathbf{Q}}(t_-, t_+) = N_{\lambda\mathbf{Q}}^<(t, t). \tag{9}$$

It is useful to write the two-particle Green's function according to

$$G_2(iz_i, jz_j; mz_m, nz_n) = G_{im}(z_i, z_m)G_{jn}(z_j, z_n) - L(iz_i, jz_j; mz_m, nz_n), \tag{10}$$

where $G_{im}(z_i, z_m) = -i\langle \mathcal{T}\{\hat{d}_i(z_i)\hat{d}_m^\dagger(z_m)\}\rangle$ is the one-particle Green's function and $L$ is the so called exchange-correlation (xc) function [64], which can be determined diagrammatically from the solution of the Bethe-Salpeter equation. Inserting this expression in Eq. (7) we obtain

$$N_{cv\mathbf{k},c'v'\mathbf{k}'}^{\mathbf{Q}}(z, z') = \delta_{\mathbf{Q},0}\rho_{cv\mathbf{k}}(t)\rho_{v'c'\mathbf{k}'}(t') + N_{cv\mathbf{k},c'v'\mathbf{k}'}^{\text{inc},\mathbf{Q}}(z, z'), \tag{11}$$

where

$$\rho_{\mu v\mathbf{k}}(t) \equiv -iG_{\mu v\mathbf{k}}(z, z^+) = \langle \hat{d}_{v\mathbf{k}}^\dagger(z)\hat{d}_{\mu\mathbf{k}}(z)\rangle \tag{12}$$

is the one-particle density matrix, and hence

$$N_{cv\mathbf{k},c'v'\mathbf{k}'}^{\text{inc},\mathbf{Q}}(z, z') = L_{cv\mathbf{k},c'v'\mathbf{k}'}^{\mathbf{Q}}(z, z'; z^+, z'^+) \equiv L(c\mathbf{k}+\mathbf{Q}z, v'\mathbf{k}'z'; v\mathbf{k}z^+, c'\mathbf{k}'+\mathbf{Q}z'^+). \tag{13}$$

Inserting Eq. (11) into Eq. (8), the exciton Green's function reads

$$N_{\lambda\mathbf{Q}}(z, z') = \delta_{\mathbf{Q},0}|\rho_\lambda(t)|^2 + N_{\lambda\mathbf{Q}}^{\text{inc}}(z, z'), \tag{14}$$

where the *exciton polarization* is defined according to

$$\rho_\lambda(t) \equiv \sum_{cv\mathbf{k}} A_{cv\mathbf{k}}^{\lambda 0*}\rho_{cv\mathbf{k}}(t) = \langle \hat{X}_{\lambda 0}(t)\rangle, \tag{15}$$

and the incoherent exciton Green's function $N_{\lambda\mathbf{Q}}^{\text{inc}}(z, z')$ is defined as in Eq. (8) with $N \to N^{\text{inc}}$. As the exciton wavefunction is normalized to unity, the exciton polarization scales like $\sqrt{\mathcal{N}_{\mathbf{k}}}$, as it should be. Equation (14) shows that the number of excitons, see Eq. (9), is naturally written as the sum of the number of *coherent excitons* $\delta_{\mathbf{Q},0}|\rho_\lambda(t)|^2$ and the number of *incoherent excitons* $N_{\lambda\mathbf{Q}}^{\text{inc}}(t) \equiv N_{\lambda\mathbf{Q}}^{\text{inc}<}(t, t)$, i.e.,

$$N_{\lambda\mathbf{Q}}(t) = \delta_{\mathbf{Q},0}|\rho_\lambda(t)|^2 + N_{\lambda\mathbf{Q}}^{\text{inc}}(t). \tag{16}$$

Coherent excitons are generated by optical excitations and therefore have zero momentum.

## 2.2  Ab initio Hamiltonian

The dynamics of any quasiparticle in a crystal, including the exciton, is governed by the ab initio Hamiltonian for electrons and phonons

$$\hat{H}(t) = \hat{H}_{\text{crystal}} + \hat{H}_{\text{drive}}(t). \tag{17}$$

The unperturbed crystal Hamiltonian is written in terms of normal mode displacement $\hat{U}_{\alpha\mathbf{Q}}$ and momentum $\hat{P}_{\alpha\mathbf{Q}}$ operators of the nuclear lattice, satisfying the commutation relations $[\hat{U}_{\alpha\mathbf{Q}}, \hat{P}^{\dagger}_{\alpha'\mathbf{Q}'}] = \delta_{\mathbf{Q},\mathbf{Q}'}\delta_{\alpha\alpha'}$, as well as electronic field operators $\hat{d}_{\mu\mathbf{k}}$. We have [60]

$$\hat{H}_{\text{crystal}} = \hat{H}_{0,e} + \hat{H}_{0,ph} + \hat{H}_{e-e} + \hat{H}_{e-ph}, \tag{18}$$

where

$$\hat{H}_{0,e} = \sum_{\mathbf{k}\mu\mu'} h_{\mu\mu'}(\mathbf{k})\hat{d}^{\dagger}_{\mu\mathbf{k}}\hat{d}_{\mu'\mathbf{k}}, \tag{19a}$$

$$\hat{H}_{0,ph} = \frac{1}{2}\sum_{\mathbf{Q}\alpha\alpha'} (\hat{U}^{\dagger}_{\alpha\mathbf{Q}}, \hat{P}^{\dagger}_{\alpha\mathbf{Q}}) \begin{pmatrix} \kappa_{\alpha\alpha'}(\mathbf{Q}) & 0 \\ 0 & \delta_{\alpha\alpha'} \end{pmatrix} \begin{pmatrix} \hat{U}_{\alpha'\mathbf{Q}} \\ \hat{P}_{\alpha'\mathbf{Q}} \end{pmatrix} - \sum_{\mathbf{k}\mu\mu'}\sum_{\alpha} \rho^{\text{eq}}_{\mu'\mu\mathbf{k}} g^{\mu\mu'}_{\alpha\mathbf{0}}(\mathbf{k})\hat{U}_{\alpha\mathbf{0}}, \tag{19b}$$

$$\hat{H}_{e-e} = \frac{1}{2}\sum_{\substack{\mathbf{k}\mathbf{k}'\mathbf{Q} \\ \mu\mu'\nu\nu'}} v_{\mu\mathbf{k}+\mathbf{Q}\,\nu'\mathbf{k}'-\mathbf{Q}\,\nu\mathbf{k}'\,\mu'\mathbf{k}}\hat{d}^{\dagger}_{\mu\mathbf{k}+\mathbf{Q}}\hat{d}^{\dagger}_{\nu'\mathbf{k}'-\mathbf{Q}}\hat{d}_{\nu\mathbf{k}'}\hat{d}_{\mu'\mathbf{k}}, \tag{19c}$$

$$\hat{H}_{e-ph} = \sum_{\mathbf{k}\mu\mu'}\sum_{\mathbf{Q}\alpha} \hat{d}^{\dagger}_{\mu\mathbf{k}}\hat{d}_{\mu'\mathbf{k}-\mathbf{Q}} g^{\mu\mu'}_{\alpha-\mathbf{Q}}(\mathbf{k})\hat{U}_{\alpha\mathbf{Q}}. \tag{19d}$$

In Eq. (19a), $h_{\mu\mu'}(\mathbf{k}) = \langle\mu\mathbf{k}|\frac{\hat{\mathbf{p}}^2}{2} + V(\hat{\mathbf{r}})|\mu'\mathbf{k}\rangle$ is the matrix element of the one-electron Hamiltonian, $V(\mathbf{r})$ being the potential generated by the nuclei in their equilibrium positions. Equation (19b) is the Hamiltonian of the bare phonons, with $\kappa_{\alpha\alpha'}(\mathbf{Q})$ the elastic tensor, $\rho^{\text{eq}}_{\mu'\mu\mathbf{k}} = \langle\hat{d}^{\dagger}_{\mu\mathbf{k}}\hat{d}_{\mu'\mathbf{k}}\rangle$ the equilibrium density matrix, and

$$g^{\mu\mu'}_{\alpha-\mathbf{Q}}(\mathbf{k}) = \langle\mu\mathbf{k}| \left.\frac{\partial V(\hat{\mathbf{r}})}{\partial U_{\alpha\mathbf{Q}}}\right|_{\mathbf{U}_{\alpha\mathbf{Q}}=0} |\mu'\mathbf{k}-\mathbf{Q}\rangle = g^{\mu'\mu*}_{\alpha\mathbf{Q}}(\mathbf{k}-\mathbf{Q}) \tag{20}$$

the *bare e-ph* couplings. Notice that Eq. (20) scales like $1/\sqrt{\mathcal{N}_{\mathbf{k}}}$ since the commutation relation between $\hat{U}_{\alpha\mathbf{Q}}$ and $\hat{P}^{\dagger}_{\alpha'\mathbf{Q}'}$ has been normalized to a Kronecker delta. The second term in Eq. (19b) guarantees that the time-derivatives of the nuclear momenta vanish in equilibrium [60]. The elastic tensor satisfies the exact identity (in the basis of the Born-Oppenheimer normal modes) [60, 65, 66]

$$\kappa_{\alpha\alpha'}(\mathbf{Q}) + \Pi^{\text{ph}}_{\mathbf{Q}\alpha\alpha'}(\omega = 0) = \delta_{\alpha\alpha'}\omega^2_{\alpha\mathbf{Q}}, \tag{21}$$

where $\Pi^{\text{ph}}_{\alpha\mathbf{Q}}(\omega)$ is the equilibrium phononic self-energy in the clamped-nuclei approximation, and $\omega^2_{\alpha\mathbf{Q}}$ are the eigenvalues of the Hessian of the Born-Oppenheimer energy. The *e-e* and *e-ph* interactions are described by Eqs. (19c) and (19d) respectively. The full ab initio Hamiltonian can alternatively be written in terms of the Born-Oppenheimer phononic operators $\hat{b}_{\alpha\mathbf{Q}}$ using the relations

$$\hat{U}_{\alpha\mathbf{Q}} = \frac{1}{\sqrt{2\omega_{\alpha\mathbf{Q}}}}(\hat{b}_{\alpha\mathbf{Q}} + \hat{b}^{\dagger}_{\alpha-\mathbf{Q}}) \quad , \quad \hat{P}_{\alpha\mathbf{Q}} = -i\sqrt{\frac{\omega_{\alpha\mathbf{Q}}}{2}}(\hat{b}_{\alpha\mathbf{Q}} - \hat{b}^{\dagger}_{\alpha-\mathbf{Q}}). \tag{22}$$

The driving Hamiltonian accounts for the light-matter interaction and reads

$$\hat{H}_{\text{drive}} = \sum_{\mu\nu\mathbf{k}} \Omega_{\mu\nu\mathbf{k}}(t)\hat{d}^{\dagger}_{\mu\mathbf{k}}\hat{d}_{\nu\mathbf{k}}, \tag{23}$$

where

$$\Omega_{\mu\nu\mathbf{k}}(t) \equiv \frac{1}{2c}\langle\mu\mathbf{k}|\hat{\mathbf{p}}\cdot\mathbf{A}(\hat{\mathbf{r}},t)+\mathbf{A}(\hat{\mathbf{r}},t)\cdot\hat{\mathbf{p}}+\frac{1}{c}\mathbf{A}^2(\hat{\mathbf{r}},t)|\nu\mathbf{k}\rangle \tag{24}$$

can be thought of as a time-dependent Rabi frequency, $\mathbf{A}$ being the vector potential.

## 2.3 The exchange-correlation function

In this section we introduce the basic mathematical tools for the development of the theory. The *exact* xc function $L^{\mathbf{Q}}_{\mu\nu\mathbf{k},\mu'\nu'\mathbf{k}'}$, see Eq. (10), satisfies the Bethe-Salpeter equation [60]

$$L = \widetilde{L} - i\widetilde{L}(v + gD_0g)L, \tag{25}$$

which can be represented diagrammatically as

$$\tag{26}$$

In Eq. (25) we have the bare *e-e* coupling $v$ (wiggly line), the bare *e-ph* coupling $g$ (circle) and the noninteracting phonon Green's function $D_0$ (spring). We recall that $D_0$ does not have poles at the physical phonon frequencies [60]. The latter emerge when dressing the phonons with electrons, see again Eq. (21). The correlator $\widetilde{L}$ is irreducible with respect to a cut of an *e-e* interaction line and/or a phonon line.

Let us introduce the $v$-reducible and $D$-irreducible correlator

$$L^{(v)} \equiv \widetilde{L} - i\widetilde{L}vL^{(v)}. \tag{27}$$

Iterating Eq. (25), and grouping terms with the same power of $g$, we get the *exact* identity

$$\begin{aligned}
L &= (\widetilde{L} - i\widetilde{L}v\widetilde{L} + \ldots) - i(\widetilde{L} - i\widetilde{L}v\widetilde{L} + \ldots)gD_0g(\widetilde{L} - i\widetilde{L}v\widetilde{L} + \ldots) \\
&\quad - i(\widetilde{L} - i\widetilde{L}v\widetilde{L} + \ldots)gD_0g(-i)(\widetilde{L} - i\widetilde{L}v\widetilde{L} + \ldots)gD_0g(\widetilde{L} - i\widetilde{L}v\widetilde{L} + \ldots) + \ldots \\
&= L^{(v)} - i\widetilde{L}g^sD_0g^s\widetilde{L} - i\widetilde{L}g^sD_0g(-i)\widetilde{L}g^sD_0g^s\widetilde{L} + \ldots \\
&= L^{(v)} - i\widetilde{L}g^s\big(D_0 + D_0(-i)g\widetilde{L}g^sD_0 + \ldots\big)g^s\widetilde{L} \\
&= L^{(v)} - i\widetilde{L}g^sDg^s\widetilde{L}. 
\end{aligned} \tag{28}$$

In the second equality we have recognized the *screened e-ph* coupling $g^s = (1 - ivL^{(v)})g = (1 - iW\widetilde{L})g$, defined diagrammatically by the equation below [60, 65]:

$$\tag{29}$$

where the circle is $g$, the crossed circle is $g^s$ and the double wiggly line is the screened Coulomb interaction $W = v - iv\widetilde{L}W$. In the third equality we have recognized the phononic self-energy $\Pi^{\text{ph}} = -ig\widetilde{L}g^s$ [60, 65], which, in the forth equality, has been used to dress the phonon Green's

function $D = D_0 + D_0 \Pi^{\text{ph}} D$. Taking into account that $g^s \widetilde{L} = g L^{(\nu)}$, the xc function can alternatively be written as

$$L = L^{(\nu)} - iL^{(\nu)} g D g L^{(\nu)} = L^{(\nu)} - i\widetilde{L} g^s D g L^{(\nu)} = L^{(\nu)} - iL^{(\nu)} g D g^s \widetilde{L}. \tag{30}$$

Henceforth we refer to the two-time functions

$$\widetilde{N}^{\mathbf{Q}}_{c\nu\mathbf{k},c'\nu'\mathbf{k}'}(z,z') = \widetilde{L}^{\mathbf{Q}}_{c\nu\mathbf{k},c'\nu'\mathbf{k}'}(z,z';z^+,z'^+), \tag{31}$$

as the irreducible exciton propagator,

$$N^{(\nu)\mathbf{Q}}_{c\nu\mathbf{k},c'\nu'\mathbf{k}'}(z,z') = L^{(\nu)\mathbf{Q}}_{c\nu\mathbf{k},c'\nu'\mathbf{k}'}(z,z';z^+,z'^+), \tag{32}$$

as the $\nu$-reducible exciton propagator, and

$$\widetilde{N}^{(D)\mathbf{Q}}_{c\nu\mathbf{k},c'\nu'\mathbf{k}'}(z,z') = -i[\widetilde{L} g^s D g^s \widetilde{L}]^{\mathbf{Q}}_{c\nu\mathbf{k},c'\nu'\mathbf{k}'}(z,z';z^+,z'^+), \tag{33}$$

as the $D$-reducible exciton propagator.

Excitonic effects in photoabsorption spectra are captured by the following approximation:

$$\widetilde{L} \simeq \widetilde{L}^{\text{SEX}} = \ell + i\ell K^{(r),\text{SEX}} \ell, \tag{34}$$

which can be represented diagrammatically as

$$\tag{35}$$

In this equation $\ell = GG$ is the free electron-hole propagator, whereas $K^{(r),\text{SEX}}$ is the $\ell$-reducible SEX kernel. The latter solves the $T$-matrix equation

$$K^{(r),\text{SEX}} = W + iW\ell K^{(r),\text{SEX}}, \tag{36}$$

or, diagrammatically,

$$\tag{37}$$

where $W$ is the statically screened Coulomb interaction. Inserting Eq. (36) into Eq. (34) we can write $\widetilde{L}^{\text{SEX}} = \ell + i\ell W \widetilde{L}^{\text{SEX}}$, which corresponds to approximate $L^{(\nu)}$ in Eq. (27) as

$$L^{(\nu)} \simeq L^{\text{HSEX}} = \ell + i\ell K^{\text{HSEX}} L^{\text{HSEX}} = \ell + i\ell K^{(r),\text{HSEX}} \ell, \tag{38}$$

with $K^{(r),\text{HSEX}} = K^{\text{HSEX}} + iK^{\text{HSEX}} \ell K^{(r),\text{HSEX}}$ the $\ell$-reducible HSEX kernel, and $K^{\text{HSEX}} = W - \nu$ the $\ell$-irreducible HSEX kernel defined in Eq. (3). The diagrammatic representation of $K^{(r)\text{HSEX}}$ is given by the equation below

$$\tag{39}$$

Solving Eq. (38) in equilibrium is equivalent to solving Eq. (2). In fact, the so called Bethe-Salpeter Hamiltonian is nothing but the "pole" of (the retarded) $L^{\text{HSEX}}$. Generally speaking, elevating the pole of a quasiparticle correlator (in our case, the exciton) to the status of a Hamiltonian is not recommended when the quasiparticles interact with other degrees of freedom (such as the phonons in our case).

Denoting by $N^{\text{HSEX}}$ the two-time propagator in Eq. (13) evaluated with $L = L^{\text{HSEX}}$, we have

$$N^{\text{HSEX},\mathbf{Q}}_{c v \mathbf{k}, c' v' \mathbf{k}'}(z, z') = \sum_{\lambda} A^{\lambda \mathbf{Q}}_{c v \mathbf{k}} N^{\text{HSEX}}_{\lambda \mathbf{Q}}(z, z') A^{\lambda \mathbf{Q}*}_{c' v' \mathbf{k}'}, \tag{40}$$

where, for systems in equilibrium,

$$N^{\text{HSEX}}_{\lambda \mathbf{Q}}(z, z') = \theta(z, z') e^{-i E_{\lambda \mathbf{Q}}(t - t')}. \tag{41}$$

In $N^{\text{HSEX}}$ the particle-hole scatters through the direct (with bare $v$) and exchange (with screened $W$) channels. Out of equilibrium $N^{\text{HSEX},<}_{\lambda \mathbf{Q}}(t, t) = N^{\text{HSEX}}(t_-, t_+)$ scales like the square of the excitation density. We therefore discard its contribution to the number of incoherent excitons, see Eq. (9).

## 3  Coherent excitons

The equation of motion for the exciton polarization $\rho_\lambda(t)$ follows from the equation of motion of $\rho_{c v \mathbf{k}}(t)$ [67]:

$$\frac{d}{dt} \rho_{c v \mathbf{k}} + i \big( \epsilon_{c \mathbf{k}} - \epsilon_{v \mathbf{k}} - i \Gamma^{\text{pol}}_{c v \mathbf{k}} \big) \rho_{c v \mathbf{k}} - i \sum_{c' v' \mathbf{k}'} K^{\text{HSEX},\mathbf{0}}_{c v \mathbf{k}, c' v' \mathbf{k}'} \rho_{c' v' \mathbf{k}'} = -i \Omega_{c v \mathbf{k}}. \tag{42}$$

In this equation $\Gamma^{\text{pol}}_{c v \mathbf{k}}$ are the polarization rates, which can be calculated by different means [44, 68, 69], although they are often treated as fitting parameters. For small excitation densities $\Gamma^{\text{pol}}_{c v \mathbf{k}}$ is dominated by *e-ph* scattering mechanisms [32, 38]. The polarization rates in the Fan-Migdal approximation can be obtained using the mirrored form of the Generalized Kadanoff-Baym Ansatz (MGKBA) [67, 70] (the MGKBA corrects the standard GKBA which leads to unphysical polarization rates).

The treatment of Ref. [67] must be improved for semiconductors hosting excitons. The origin of the term $\Gamma^{\text{pol}}_{c v \mathbf{k}} \rho_{c v \mathbf{k}}$ stems from the collision integral

$$\Gamma^{\text{pol}}_{c v \mathbf{k}}(t) \rho_{c v \mathbf{k}}(t) = S_{c v \mathbf{k}}(t) = \int d\bar{z} \big[ \Sigma(z, \bar{z}) G(\bar{z}, z^+) - G(z, \bar{z}) \Sigma(\bar{z}, z^+) \big]_{c v \mathbf{k}}, \tag{43}$$

where $\Sigma$ is the electronic correlation energy. In the presence of excitons, the Fan-Migdal self-energy alone (see the first diagram in Fig. 1) is not sufficient because electrons and holes cannot form bound states. A suitable approximation for $\Sigma$ can be deduced from those approaches that treat excitons as composite bosonic particles [69]. In these approaches the *exciton* self-energy has the structure $\Pi^{\text{ex}}(z, z') = i \mathcal{G} D(z, z') N^{\text{HSEX}}(z, z') \mathcal{G}$, where the *exciton-phonon coupling* [42, 53, 71–73]

$$\mathcal{G}^{\lambda \lambda'}_{\alpha - \mathbf{Q}'}(\mathbf{Q}) \equiv \sum_{c_1 c_2 v_1 \mathbf{k}_1} A^{\lambda \mathbf{Q}*}_{c_1 v_1 \mathbf{k}_1} g^{s, c_1 c_2}_{\alpha - \mathbf{Q}'}(\mathbf{k}_1 + \mathbf{Q}) A^{\lambda' \mathbf{Q} - \mathbf{Q}'}_{c_2 v_1 \mathbf{k}_1} - \sum_{c_1 v_1 v_2 \mathbf{k}_1} A^{\lambda \mathbf{Q}*}_{c_1 v_1 \mathbf{k}_1} g^{s, v_2 v_1}_{\alpha - \mathbf{Q}'}(\mathbf{k}_1 + \mathbf{Q}') A^{\lambda' \mathbf{Q} - \mathbf{Q}'}_{c_1 v_2 \mathbf{k}_1 + \mathbf{Q}'} \tag{44}$$

depends on the *screened e-ph* coupling $g^s$ and the exciton wavefunctions defined in Eq. (2).

In the ab initio formulation, the polarization rates generated by the model $\Pi^{\text{ex}}$ are produced by the *electronic* self-energy in Fig. 1(a), see below for the proof of this statement. However,

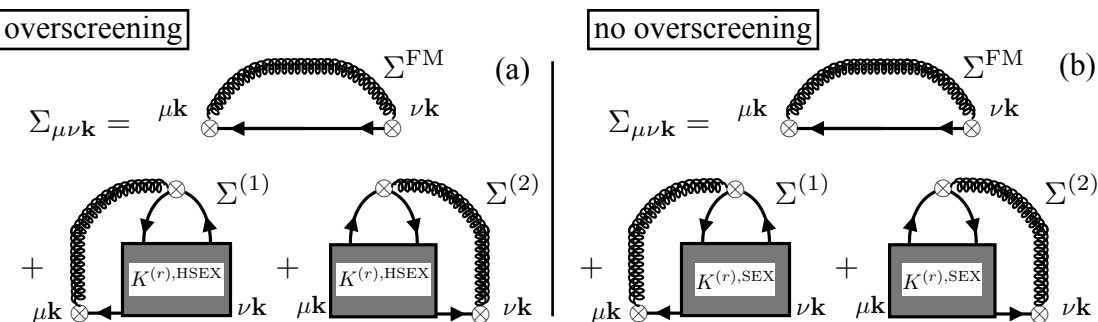

Figure 1: Electronic self-energy with excitonic effects in terms of electronic Green's functions $G$ (solid lines), phononic Green's functions $D$ (double springs), and screened $e$-$ph$ couplings $g^s$ (circled crosses). Panel (a) shows $\Sigma$ affected by overscreening. Panel (b) shows $\Sigma$ with no overscreening.

the screened $e$-$ph$ coupling in Fig. 1(a) leads to an overscreening. Consider, for instance, the second diagram $\Sigma^{(1)}$. Taking into account that $W = v - i v \widetilde{L}^{\text{SEX}} W$, and using the explicit form of $g^s$ for the $e$-$ph$ coupling at the top of the diagram (with $\widetilde{L} \simeq \widetilde{L}^{\text{SEX}}$), we get the diagrammatic structure

$$
\begin{aligned}
g^s \ell K^{(r),\text{HSEX}} &= g \ell K^{(r),\text{HSEX}} - i g \widetilde{L}^{\text{SEX}} W \ell K^{(r),\text{HSEX}} \\
&= g \ell K^{(r),\text{HSEX}} - i g \widetilde{L}^{\text{SEX}} (v - i v \widetilde{L}^{\text{SEX}} v + \ldots) \ell (K^{\text{HSEX}} + i K^{\text{HSEX}} \ell K^{\text{HSEX}} + \ldots) \\
&= g \ell K^{(r),\text{HSEX}} - i g \widetilde{L}^{\text{SEX}} (v - i v \widetilde{L}^{\text{SEX}} v + \ldots) L^{\text{HSEX}} K^{\text{HSEX}} \\
&= g \ell K^{(r),\text{HSEX}} - i g \widetilde{L}^{\text{SEX}} (v - i v \widetilde{L}^{\text{SEX}} v + \ldots)(\widetilde{L}^{\text{SEX}} - i \widetilde{L}^{\text{SEX}} v \widetilde{L}^{\text{SEX}} + \ldots) K^{\text{HSEX}}, \quad (45)
\end{aligned}
$$

from which the overscreening of the $e$-$ph$ coupling is evident.

To solve the overscreening problem and simultaneously develop a theory in terms of screened $e$-$ph$ couplings, it is sufficient to examine the exact form of the electronic self-energy [60,65]. The closest approximation to Fig. 1(a) which is free of overscreening is the one in Fig. 1(b), where $K^{(r),\text{HSEX}} \to K^{(r),\text{SEX}}$. The crucial difference between the self-energies in Fig. 1 is that $\Sigma G \propto g^s D g^s L^{\text{HSEX}}$ for panel (a), see Eq. (38), while $\Sigma G \propto g^s D g^s \widetilde{L}^{\text{SEX}} = g^s D g L^{\text{HSEX}}$ for panel (b), see Eq. (34). To evaluate the collision integral, see Eq. (43), with the self-energy of panel (b) we make the following ansatz for the off-diagonal elements of the electronic Green's function [74]

$$
G_{cv\mathbf{k}}(z,z') = -G_{cc\mathbf{k}}(z,z')\rho_{cv\mathbf{k}}(t') + \rho_{cv\mathbf{k}}(t)G_{vv\mathbf{k}}(z,z'), \tag{46a}
$$

$$
G_{vc\mathbf{k}}(z,z') = -\rho_{vc\mathbf{k}}(t)G_{cc\mathbf{k}}(z,z') + G_{vv\mathbf{k}}(z,z')\rho_{vc\mathbf{k}}(t'). \tag{46b}
$$

This ansatz is exact at the mean-field level and/or for small excitation densities. Equation (46) allows for expressing all matrix elements of the irreducible xc function $\widetilde{L}^{\text{SEX}}$ in terms of the irreducible exciton propagator [compare with Eq. (13)]

$$
\widetilde{N}^{\text{SEX},\mathbf{Q}}_{cv\mathbf{k},c'v'\mathbf{k}'}(z,z') = \widetilde{L}^{\text{SEX},\mathbf{Q}}_{cv\mathbf{k},c'v'\mathbf{k}'}(z,z';z^+,z'^+), \tag{47}
$$

see Appendix A for details. The irreducible exciton propagator takes a particularly simple form in the irreducible excitonic basis. Let us consider the eigenvalue equation [compare with Eq. (2)]

$$
E^{\mathbf{Q}}_{cv\mathbf{k}} \widetilde{A}^{\widetilde{\lambda}\mathbf{Q}}_{cv\mathbf{k}} - \sum_{c'v'\mathbf{k}'} W^{\mathbf{Q}}_{cv\mathbf{k},c'v'\mathbf{k}'} \widetilde{A}^{\widetilde{\lambda}\mathbf{Q}}_{c'v'\mathbf{k}'} = \widetilde{E}_{\widetilde{\lambda}\mathbf{Q}} \widetilde{A}^{\widetilde{\lambda}\mathbf{Q}}_{cv\mathbf{k}}. \tag{48}
$$

The irreducible exciton wavefunctions $\widetilde{A}$ are chosen orthonormal for every $\mathbf{Q}$. Proceeding along the same lines leading to Eq. (40) we obtain

$$\widetilde{N}^{\text{SEX},\mathbf{Q}}_{c v \mathbf{k}, c' v' \mathbf{k}'}(z, z') = \sum_{\widetilde{\lambda}} \widetilde{A}^{\widetilde{\lambda}\mathbf{Q}}_{c v \mathbf{k}} \widetilde{N}^{\text{SEX}}_{\widetilde{\lambda}\mathbf{Q}}(z, z') \widetilde{A}^{\widetilde{\lambda}\mathbf{Q}*}_{c' v' \mathbf{k}'}, \tag{49}$$

where we approximate $N^{\text{HSEX}}$ with its equilibrium expression, i.e.,

$$\widetilde{N}^{\text{SEX}}_{\widetilde{\lambda}\mathbf{Q}}(z, z') = \theta(z, z') e^{-i\widetilde{E}_{\widetilde{\lambda}\mathbf{Q}}(t - t')}. \tag{50}$$

These results can be used to rewrite the collision integral in the excitonic basis according to

$$S_\lambda(z) \equiv \sum_{c v \mathbf{k}} A^{\lambda\mathbf{0}*}_{c v \mathbf{k}} S_{c v \mathbf{k}}(z) = i \sum_{\lambda_1 \lambda_2} \sum_{\alpha \mathbf{Q}} \int d\bar{z} \, \widetilde{\mathcal{G}}^{\widetilde{\lambda}_1 \lambda *}_\alpha(\mathbf{Q}) \widetilde{N}^{\text{SEX}}_{\widetilde{\lambda}_1 \mathbf{Q}}(z, \bar{z}) \widetilde{\mathcal{G}}^{\widetilde{\lambda}_1 \lambda_2}_\alpha(\mathbf{Q}) \rho_{\lambda_2}(\bar{t}) D_{\alpha - \mathbf{Q}}(z, \bar{z}), \tag{51}$$

where we define the *irreducible exciton-phonon coupling*

$$\widetilde{\mathcal{G}}^{\widetilde{\lambda}\lambda'}_\alpha(\mathbf{Q}) \equiv \sum_{c_1 c_2 v_1 \mathbf{k}_1} \widetilde{A}^{\widetilde{\lambda}\mathbf{Q}*}_{c_1 v_1 \mathbf{k}_1} g^{s,c_1 c_2}_{\alpha - \mathbf{Q}}(\mathbf{k}_1 + \mathbf{Q}) A^{\lambda' \mathbf{0}}_{c_2 v_1 \mathbf{k}_1} - \sum_{c_1 v_1 v_2 \mathbf{k}_1} \widetilde{A}^{\widetilde{\lambda}\mathbf{Q}*}_{c_1 v_1 \mathbf{k}_1} g^{s,v_2 v_1}_{\alpha - \mathbf{Q}}(\mathbf{k}_1 + \mathbf{Q}) A^{\lambda' \mathbf{0}}_{c_1 v_2 \mathbf{k}_1 + \mathbf{Q}}. \tag{52}$$

Equation (52) does not reduce to Eq. (44) with $\mathbf{Q}' = \mathbf{Q}$ as the *e-ph* coupling is contracted with the product of an exciton wavefunction $A$ and an irreducible exciton wavefunction $\widetilde{A}$. Assuming that the dominant contribution in Eq. (51) comes from the terms with $\lambda_2 = \lambda$, we show in Appendix A that the Markov approximation enables us to express Eq. (42) in the excitonic basis as follows

$$\frac{d}{dt}\rho_\lambda(t) = -iE_{\lambda\mathbf{0}}\rho_\lambda(t) - i\Omega_\lambda(t) - \frac{1}{2}\sum_{\widetilde{\lambda}'\mathbf{Q}} \widetilde{\Gamma}^{\text{pol}}_{\lambda\widetilde{\lambda}'\mathbf{Q}} \rho_\lambda(t), \tag{53}$$

with $\Omega_\lambda \equiv \sum_{c v k} A^{\lambda\mathbf{0}*}_{c v \mathbf{k}} \Omega_{c v \mathbf{k}}$. The polarization rates depend on the phononic populations $f^{\text{ph}}_{\alpha\mathbf{Q}} \equiv \langle \hat{b}^\dagger_{\alpha\mathbf{Q}} \hat{b}_{\alpha\mathbf{Q}} \rangle$ according to

$$\widetilde{\Gamma}^{\text{pol}}_{\lambda\widetilde{\lambda}'\mathbf{Q}} = 2\pi \sum_\alpha \frac{|\widetilde{\mathcal{G}}^{\widetilde{\lambda}'\lambda}_\alpha(\mathbf{Q})|^2}{2\omega_{\alpha\mathbf{Q}}} \Big[ \delta(\widetilde{E}_{\widetilde{\lambda}'\mathbf{Q}} - E_{\lambda\mathbf{0}} + \omega_{\alpha\mathbf{Q}})(1 + f^{\text{ph}}_{\alpha - \mathbf{Q}}) + \delta(\widetilde{E}_{\widetilde{\lambda}'\mathbf{Q}} - E_{\lambda\mathbf{0}} - \omega_{\alpha\mathbf{Q}}) f^{\text{ph}}_{\alpha\mathbf{Q}} \Big]. \tag{54}$$

We remark that $\widetilde{\Gamma}^{\text{pol}}$ involves the difference between an exciton energy $E$ and an irreducible exciton energy $\widetilde{E}$. The necessity of introducing irreducible excitons in a theory of excitons and phonons has also been recognized by Paleari and Marini [50]. They investigated a scattering channel different from the one considered here. Nevertheless, their study also led to the emergence of the irreducible exciton-phonon coupling in Eq. (52).

The equation of motion Eq. (53) for the exciton polarization agrees with cluster expansion results [12, 53, 54] provided that we replace $\widetilde{\mathcal{G}}$ with $\mathcal{G}$ and $\widetilde{E}_{\widetilde{\lambda}'\mathbf{Q}}$ with $E_{\lambda'\mathbf{Q}}$. Such replacement is equivalent to evaluate the electronic self-energy with the diagrams of Fig. 1(a) instead of Fig. 1(b). As previously discussed, this introduces an overscreening of the *e-ph* coupling. The overscreening issue is common to all methods that do not rely on the ab initio Hamiltonian [60], but rather on model Hamiltonians where both the *e-e* and *e-ph* interactions are already screened. Our treatment highlights the advantages of the first-principles Green's function formulation, where screening naturally emerges from the diagrammatic expansion and is therefore counted only once. Another important feature of Eq. (53) is that it reduces to the equation of motion in Ref. [67] if we neglect the last two diagrams in Fig. 1.

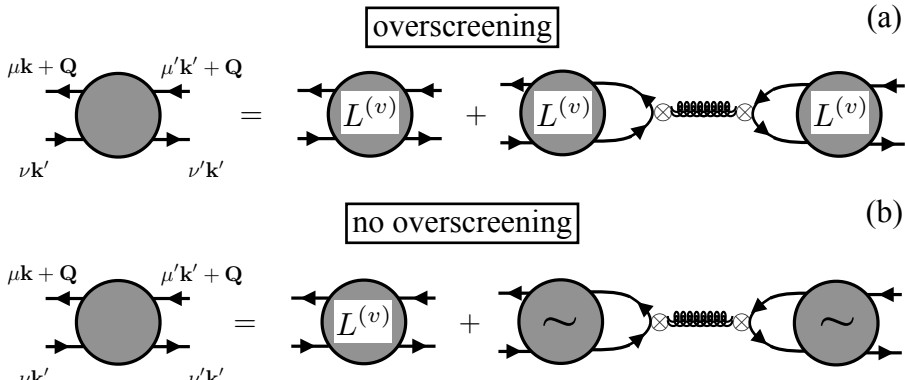

Figure 2: Exchange correlation function $L$ underlying the equation of motion of the number of incoherent excitons. In panel (a) we show $L$ from model Hamiltonians whereas in panel (b) we show the exact result from the ab initio Hamiltonian.

As a technical note, we highlight that there exists a generalization of Eq. (53) which agrees with more sophisticated treatments of the bosonic Hamiltonian for excitons and phonons [69], see Appendix A. The main difference with these models is again the occurrence of $\widetilde{\mathcal{G}}$ and $\widetilde{E}$. By retaining all terms with $\lambda_2 \neq \lambda$ in Eq. (51) we show that $S^\lambda = \sum_{\widetilde{\lambda}_1 \lambda_2 \mathbf{Q}} \widetilde{\Gamma}^{\text{pol}}_{\lambda \lambda_2 \widetilde{\lambda}_1 \mathbf{Q}} \rho_{\lambda_2}$ where $\widetilde{\Gamma}^{\text{pol}}_{\lambda \lambda_2 \widetilde{\lambda}_1 \mathbf{Q}}$ can be obtained from Eq. (54) by replacing $|\widetilde{\mathcal{G}}^{\widetilde{\lambda}_1 \lambda}_\alpha(\mathbf{Q})|^2 \rightarrow \widetilde{\mathcal{G}}^{\widetilde{\lambda}_1 \lambda *}_\alpha(\mathbf{Q}) \widetilde{\mathcal{G}}^{\widetilde{\lambda}_1 \lambda_2 *}_\alpha(\mathbf{Q})$ and $E_{\lambda \mathbf{0}} \rightarrow E_{\lambda_2 \mathbf{0}}$. It should be stressed, however, that the matrix $\sum_{\widetilde{\lambda}_1 \mathbf{Q}} \Gamma^{\text{coh}}_{\lambda \lambda_2 \widetilde{\lambda}_1 \mathbf{Q}}$ is not guaranteed to be positive definite.

## 4   Incoherent excitons

In this section we show how optically generated coherent excitons are converted into incoherent excitons via the *e-ph* scattering. The cluster expansion technique applied to model Hamiltonians with already screened *e-e* and *e-ph* couplings has been used to derive the equation of motion for incoherent excitons [12,53,54]. This equation can alternatively be derived using many-body Green's function methods. It is sufficient to approximate the xc function $L$ as in Fig. 2(a), see below. Using the same arguments as in the previous section, it is straightforward to see that $L$ in Fig. 2(a) suffers from overscreening. To highlight the shortcomings of Fig. 2(a), we show in Fig. 2(b) the exact result from Eq. (28).

We approximate again $\widetilde{L} \simeq \widetilde{L}^{\text{SEX}}$ and hence $L^{(v)} \simeq L^{\text{HSEX}}$. The propagator $N^{\text{HSEX}}(z, z') = L^{\text{HSEX}}(z, z'; z, z')$ does not contribute to the number of incoherent excitons, see discussion below Eq. (41). We then focus on the second term of Eq. (28), i.e., the $D$-reducible exciton propagator

$$N^{(D)} \simeq -i\widetilde{L}^{\text{SEX}} g^s D g^s \widetilde{L}^{\text{SEX}}. \tag{55}$$

As we see later, phonons are responsible for converting all coherent excitons into $D$-reducible excitons. We observe that $N^{(D)}$ involves "off-diagonal" elements of $\widetilde{L}^{\text{SEX}}$. Suppose that the left *e-ph* coupling has conduction indices. Then the left part of the diagram is $\widetilde{L}^{\text{SEX},\mathbf{Q}}_{cv\mathbf{k},c_1 c_2 \mathbf{k}_1}(z, z'; z, z')$. On the other hand, if the left *e-ph* coupling has valence indices then the left part of the diagram is $\widetilde{L}^{\text{SEX},\mathbf{Q}}_{cv\mathbf{k},v_2 v_1 \mathbf{k}_1}(z, z'; z, z')$. Analogous considerations apply to the right part of the diagram. Through the ansatz in Eqs. (46) we can express all matrix elements of $\widetilde{L}^{\text{SEX}}$ in terms of $\widetilde{N}^{\text{SEX}}$, see Appendix A. Expanding $\widetilde{N}^{\text{SEX}}$ in the irreducible excitonic basis, see Eq. (49), and $\rho$ in the

excitonic basis, see Eqs. (15), we get

$$N_{\widetilde{\lambda}\mathbf{Q}}^{(D)}(z,z') = -\int d\bar{z}d\bar{z}' \sum_{\lambda_1\lambda_2\alpha} \widetilde{\mathcal{G}}_{\alpha}^{\widetilde{\lambda}\lambda_1}(\mathbf{Q})\widetilde{\mathcal{G}}_{\alpha}^{\widetilde{\lambda}\lambda_2*}(\mathbf{Q})\widetilde{N}_{\widetilde{\lambda}\mathbf{Q}}^{\mathrm{SEX}}(z,\bar{z})\rho_{\lambda_1}(\bar{t})iD_{\alpha\mathbf{Q}}(\bar{z},\bar{z}')\rho_{\lambda_2}^*(\bar{t}')\widetilde{N}_{\widetilde{\lambda}\mathbf{Q}}^{\mathrm{SEX}}(\bar{z}',z').$$

(56)

In the following we assume that the dominant term comes from $\lambda_1 = \lambda_2$. Taking into account Eq. (50) we find

$$\Big[i\frac{d}{dz} - \widetilde{E}_{\widetilde{\lambda}\mathbf{Q}}\Big]N_{\widetilde{\lambda}\mathbf{Q}}^{(D)}(z,z') = \int d\bar{z}\,\Pi_{\widetilde{\lambda}\mathbf{Q}}^{(D)}(z,\bar{z})\widetilde{N}_{\widetilde{\lambda}\mathbf{Q}}^{\mathrm{SEX}}(\bar{z},z'),$$

(57a)

$$\Big[-i\frac{d}{dz'} - \widetilde{E}_{\widetilde{\lambda}\mathbf{Q}}\Big]N_{\widetilde{\lambda}\mathbf{Q}}^{(D)}(z,z') = \int d\bar{z}\,\widetilde{N}_{\widetilde{\lambda}\mathbf{Q}}^{\mathrm{SEX}}(z,\bar{z})\Pi_{\widetilde{\lambda}\mathbf{Q}}^{(D)}(\bar{z},z'),$$

(57b)

where the self-energy for the $D$-reducible excitons reads

$$\Pi_{\widetilde{\lambda}\mathbf{Q}}^{(D)}(z,z') = \sum_{\lambda'\alpha} |\widetilde{\mathcal{G}}_{\alpha}^{\widetilde{\lambda}\lambda'}(\mathbf{Q})|^2 \rho_{\lambda'}(t)D_{\alpha\mathbf{Q}}(z,z')\rho_{\lambda'}^*(t').$$

(58)

We extract the time-derivative of the number of $D$-reducible excitons

$$N^{(D)}(t) \equiv N^{(D),<}(t,t),$$

(59)

by taking $z = t_-$ and $z' = t'_+$, subtracting Eq. (57b) from Eq. (57a) and then setting $t = t'$:

$$\frac{d}{dt}N_{\widetilde{\lambda}\mathbf{Q}}^{(D)}(t) = i\int^t d\bar{t}\,\Big[\Pi_{\widetilde{\lambda}\mathbf{Q}}^{(D),<}(t,\bar{t})\widetilde{N}_{\widetilde{\lambda}\mathbf{Q}}^{\mathrm{SEX},>}(\bar{t},t) - \Pi_{\widetilde{\lambda}\mathbf{Q}}^{(D),>}(t,\bar{t})\widetilde{N}_{\widetilde{\lambda}\mathbf{Q}}^{\mathrm{SEX},<}(\bar{t},t)\Big] + \mathrm{h.c.},$$

(60)

where $\widetilde{N}^{\mathrm{SEX},\gtrless}(t,t')$ can be deduced from Eq. (50). To evaluate the collision integral in Eq. (60) we implement again the Markov approximation and find

$$\frac{d}{dt}N_{\widetilde{\lambda}\mathbf{Q}}^{(D)}(t) = \sum_{\lambda'} \widetilde{\Gamma}_{\lambda'\widetilde{\lambda}\mathbf{Q}}^{\mathrm{pol}} |\rho_{\lambda'}(t)|^2,$$

(61)

where the polarization rates are defined in Eqs. (54).

By definition, see Eq. (13), the number of incoherent exciton with quantum number $\lambda$ and momentum $\mathbf{Q}$ produced by our approximation is

$$N_{\lambda\mathbf{Q}}^{\mathrm{inc}}(t) = \sum_{\widetilde{\lambda}'} |S_{\lambda\widetilde{\lambda}'}^{\mathbf{Q}}|^2 N_{\widetilde{\lambda}'\mathbf{Q}}^{(D)}(t),$$

(62)

where

$$S_{\lambda\widetilde{\lambda}'}^{\mathbf{Q}} = \sum_{cv\mathbf{k}} A_{cv\mathbf{k}}^{\lambda\mathbf{Q}*}\widetilde{A}_{cv\mathbf{k}}^{\widetilde{\lambda}'\mathbf{Q}}$$

(63)

is the overlap matrix between excitons and irreducible excitons. Therefore, the equation of motion for the number of incoherent excitons reads

$$\frac{d}{dt}N_{\lambda\mathbf{Q}}^{\mathrm{inc}}(t) = \sum_{\lambda'\widetilde{\lambda}'} |S_{\lambda\widetilde{\lambda}'}^{\mathbf{Q}}|^2 \widetilde{\Gamma}_{\lambda'\widetilde{\lambda}'\mathbf{Q}}^{\mathrm{pol}} |\rho_{\lambda'}(t)|^2.$$

(64)

From the equations of motion Eqs. (53) and (64) we can easily deduce the equation of motion for the total number of excitons, see Eq. (16),

$$N = \sum_{\lambda\mathbf{Q}} N_{\lambda\mathbf{Q}} = \sum_{\lambda\mathbf{Q}} \big(\delta_{\mathbf{Q},\mathbf{0}}|\rho_\lambda|^2 + N_{\lambda\mathbf{Q}}^{\mathrm{inc}}\big).$$

(65)

Since $\sum_\lambda |S^{\mathbf{Q}}_{\lambda\widetilde{\lambda}'}|^2 = 1$ for all $\widetilde{\lambda}'$ we have $\sum_\lambda N^{\text{inc}}_{\lambda\mathbf{Q}}(t) = \sum_{\widetilde{\lambda}} N^{(D)}_{\widetilde{\lambda}\mathbf{Q}}(t)$, and therefore

$$\frac{d}{dt}N = 2\sum_\lambda \text{Im}\big[\Omega_\lambda \rho^*_\lambda\big]. \tag{66}$$

The ab initio formulation confirms an important result from model Hamiltonians, which states that the total number of excitons remains constant after the optical field is applied.

# 5 Inelastic exciton-phonon scattering

The approximations to $\Sigma$ and $L$ discussed in Sections 3 and 4 do not include the inelastic exciton-phonon scattering, responsible for exciton diffusion and thermalization. The inclusion of this fundamental process in the exciton dynamics requires an improvement of the xc function $L^{(v)}$. The exact $L^{(v)}$ satisfies the Bethe-Salpeter equation

$$L^{(v)} = L^{\text{HSEX}} + iL^{\text{HSEX}}K^{\text{c}}L^{(v)}, \tag{67}$$

where the the correlation kernel $K^{\text{c}}$ is irreducible with respect to a cut of a $v$-, $D$- and $\ell$-line. To second-order in the *screened e-ph* coupling and for small excitation densities we have

$$
\begin{aligned}
K^{\text{c},\mathbf{Q}}_{c_1 v_1 \mathbf{k}_1, c_2 v_2 \mathbf{k}_2}(z_1, z_2; z_3, z_4) = &-\delta(z_1, z_3)\delta(z_2, z_4)\sum_{\alpha \mathbf{Q}'} D_{\alpha\mathbf{Q}'}(z_1, z_2) \\
&\times\Big[\sum_{c_1' c_2'} g^{d,c_1 c_1'}_{\alpha-\mathbf{Q}'}(\mathbf{k}_1+\mathbf{Q}) g^{d,c_2' c_2}_{\alpha\mathbf{Q}'}(\mathbf{k}_2+\mathbf{Q}-\mathbf{Q}') L^{\mathbf{Q}-\mathbf{Q}'}_{c_1' v_1 \mathbf{k}_1, c_2' v_2 \mathbf{k}_2}(z_1, z_2; z_1, z_2) \\
&-\sum_{c_1' v_2'} g^{d,c_1 c_1'}_{\alpha-\mathbf{Q}'}(\mathbf{k}_1+\mathbf{Q}) g^{d,v_2 v_2'}_{\alpha\mathbf{Q}'}(\mathbf{k}_2) L^{\mathbf{Q}-\mathbf{Q}'}_{c_1' v_1 \mathbf{k}_1, c_2 v_2' \mathbf{k}_2+\mathbf{Q}'}(z_1, z_2; z_1, z_2) \\
&-\sum_{v_1' c_2'} g^{d,v_1' v_1}_{\alpha-\mathbf{Q}'}(\mathbf{k}_1+\mathbf{Q}') g^{d,c_2' c_2}_{\alpha\mathbf{Q}'}(\mathbf{k}_2+\mathbf{Q}-\mathbf{Q}') L^{\mathbf{Q}-\mathbf{Q}'}_{c_1 v_1' \mathbf{k}_1+\mathbf{Q}', c_2' v_2 \mathbf{k}_2}(z_1, z_2; z_1, z_2) \\
&+\sum_{v_1' v_2'} g^{d,v_1' v_1}_{\alpha-\mathbf{Q}'}(\mathbf{k}_1+\mathbf{Q}') g^{d,v_2 v_2'}_{\alpha\mathbf{Q}'}(\mathbf{k}_2) L^{\mathbf{Q}-\mathbf{Q}'}_{c_1 v_1' \mathbf{k}_1+\mathbf{Q}', c_2 v_2' \mathbf{k}_2+\mathbf{Q}'}(z_1, z_2; z_1, z_2)\Big], \tag{68}
\end{aligned}
$$

which is represented diagrammatically in Fig. 3. The assumption of small excitation density is equivalent to approximate the internal Green's functions as

$$G_{cc\mathbf{k}}(z, z') \simeq -i\theta(z, z')e^{-i\epsilon_{c\mathbf{k}}(t-t')}, \tag{69a}$$

$$G_{vv\mathbf{k}}(z, z') \simeq i\theta(z', z)e^{-i\epsilon_{v\mathbf{k}}(t-t')}. \tag{69b}$$

Consider the first diagram of the kernel. The structure $GGK^{\text{c},\mathbf{Q}}$ involves the calculation of

$$\int dz_1 dz_3 G_{c_1 \mathbf{k}_1+\mathbf{Q}}(z, z_1) G_{c_1' \mathbf{k}_1+\mathbf{Q}-\mathbf{Q}'}(z_1, z')\delta(z_3, z_1) G_{v_1 \mathbf{k}_1}(z', z_3) G_{v_1 \mathbf{k}_1}(z_3, z)$$

$$= iG_{v_1 \mathbf{k}_1}(z', z)\int dz_1 G_{c_1 \mathbf{k}_1+\mathbf{Q}}(z, z_1) G_{c_1' \mathbf{k}_1+\mathbf{Q}-\mathbf{Q}'}(z_1, z'), \tag{70}$$

where we use Eqs. (69). The gluing of two Green's functions allows us to recover the Feynman rules for the two-particle Green's function [64]. It is straightforward to verify that the gluing argument applies to all four diagrams of the kernel. We emphasize that no overscreening issue arises if we use the screened *e-ph* coupling in $K^{\text{c}}$. In fact, the Green's functions entering an *e-ph* vertex come from different xc functions, ensuring that screening is counted only once.

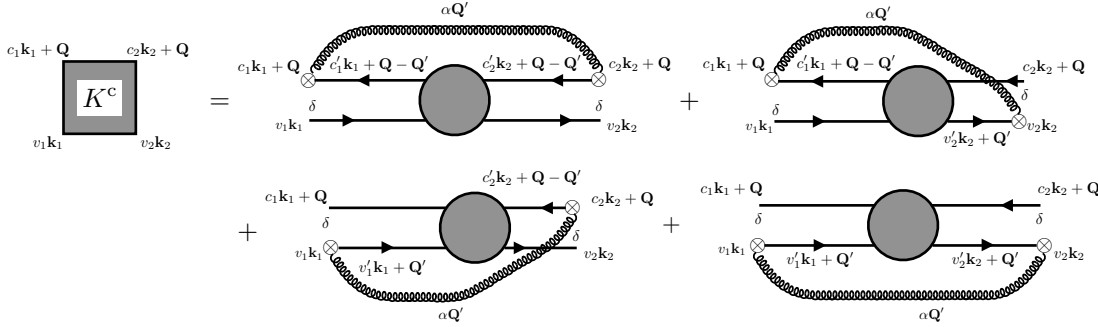

Figure 3: Diagrammatic representation of the irreducible kernel to second order in the *e-ph* interaction and for small excitation densities.

The diagrams of Fig. 3 have been discussed in Refs. [73, 75] for the stationary case and for $L = L^{\mathrm{HSEX}}$.

Since $K^{\mathrm{c}}$ is already of second order in $g^s$ we approximate [see Eq. (28)]

$$K^{\mathrm{c}} L^{(v)} = K^{\mathrm{c}} \big( L + i \widetilde{L} g^s D g^s \widetilde{L} \big) \simeq K^{\mathrm{c}} L, \tag{71}$$

which implies

$$L \simeq L^{\mathrm{HSEX}} + i L^{\mathrm{HSEX}} K^{\mathrm{c}}[L] L - i \widetilde{L} g^s D g^s \widetilde{L}. \tag{72}$$

In the following we derive the equations of motion for all kind of excitons and show how they are coupled.

## 5.1  $D$-reducible excitons

The approximation in Eq. (67) implies that, see Eq. (27),

$$\widetilde{L} = \widetilde{L}^{\mathrm{SEX}} + \widetilde{L}^{\mathrm{SEX}} K^{\mathrm{c}}[L] \widetilde{L}. \tag{73}$$

The last term in Eq. (72) with $\widetilde{L}$ from Eq. (73) provides an improved approximation to the $D$-reducible exciton propagator $N^{(D)}$, compare with Eq. (55). The equation of motion for $N^{(D),<}$ in the irreducible exciton basis is identical to Eq. (60) provided that we replace $\widetilde{N}^{\mathrm{SEX}} \to \widetilde{N}$. Due to the exciton-phonon scattering $\widetilde{N}^{<}$ is, in general, nonzero. To calculate the collision integral we extend the MGKBA to $\widetilde{N}^{\lessgtr}$ along the lines outlined Ref. [76] [compare with Eq. (50)]

$$\widetilde{N}^{\lessgtr}_{\widetilde{\lambda}\mathbf{Q}}(t, t') = \Big[ \theta(t - t') \, \widetilde{N}^{\lessgtr}_{\widetilde{\lambda}\mathbf{Q}}(t, t) + \theta(t' - t) \widetilde{N}^{\lessgtr}_{\widetilde{\lambda}\mathbf{Q}}(t', t') \Big] e^{-i \widetilde{E}_{\widetilde{\lambda}\mathbf{Q}}(t - t')}, \tag{74}$$

where $\widetilde{N}^{<}_{\widetilde{\lambda}\mathbf{Q}}(t, t) = \widetilde{N}_{\widetilde{\lambda}\mathbf{Q}}(t)$ is the number of irreducible excitons, and $\widetilde{N}^{>}_{\widetilde{\lambda}\mathbf{Q}}(t, t) = 1 + \widetilde{N}_{\widetilde{\lambda}\mathbf{Q}}(t)$. Then, the improved version of the equation of motion Eq. (61) reads

$$\frac{d}{dt} N^{(D)}_{\widetilde{\lambda}\mathbf{Q}}(t) = \sum_{\lambda'} \Big[ \widetilde{\Gamma}^{\mathrm{pol}}_{\lambda'\widetilde{\lambda}\mathbf{Q}} + \widetilde{\Gamma}_{\lambda'\widetilde{\lambda}\mathbf{Q}} \widetilde{N}_{\widetilde{\lambda}\mathbf{Q}}(t) \Big] |\rho_{\lambda'}(t)|^2, \tag{75}$$

where the rates

$$\widetilde{\Gamma}_{\lambda\widetilde{\lambda}'\mathbf{Q}} = 2\pi \sum_{\alpha} \frac{|\widetilde{\mathcal{G}}^{\widetilde{\lambda}'\lambda}_{\alpha}(\mathbf{Q})|^2}{2\omega_{\alpha\mathbf{Q}}} \Big[ \delta(\widetilde{E}_{\widetilde{\lambda}'\mathbf{Q}} - E_{\lambda\mathbf{0}} + \omega_{\alpha\mathbf{Q}}) - \delta(\widetilde{E}_{\widetilde{\lambda}'\mathbf{Q}} - E_{\lambda\mathbf{0}} - \omega_{\alpha\mathbf{Q}}) \Big] \tag{76}$$

depend on the irreducible exciton-phonon coupling defined in Eq. (52). Equation (75) couples the $D$-reducible excitons to the irreducible excitons $\widetilde{N}$ and the coherent excitons $|\rho|^2$.

## 5.2  Irreducible excitons

The equation of motion for $\widetilde{N}_{\widetilde{\lambda}\mathbf{Q}}$ can be derived from Eq. (73). As the kernel is proportional to $\delta(z_1, z_3)\delta(z_2, z_4)$ we infer that this equation can be closed on $\widetilde{N}(z, z') = \widetilde{L}(z, z'; z, z')$ and $N^{\mathrm{inc}}(z, z') = L(z, z'; z, z')$ [see Eqs. (13) and (32)], thus becoming an integral equation for two-time functions on the Keldysh contour. Considering only the diagonal elements of $\widetilde{N}$ in the irreducible exciton basis and the diagonal elements of $N^{\mathrm{inc}}$ in the exciton basis we find

$$\left[i\frac{d}{dz} - \widetilde{E}_{\widetilde{\lambda}\mathbf{Q}}\right]\widetilde{N}_{\widetilde{\lambda}\mathbf{Q}}(z, z') = i\delta(z, z') + \int d\bar{z}\, \widetilde{\Pi}_{\widetilde{\lambda}\mathbf{Q}}(z, \bar{z})\widetilde{N}_{\widetilde{\lambda}\mathbf{Q}}(\bar{z}, z'), \tag{77a}$$

$$\left[-i\frac{d}{dz'} - \widetilde{E}_{\widetilde{\lambda}\mathbf{Q}}\right]\widetilde{N}_{\widetilde{\lambda}\mathbf{Q}}(z, z') = i\delta(z, z') + \int d\bar{z}\, \widetilde{N}_{\widetilde{\lambda}\mathbf{Q}}(z, \bar{z})\widetilde{\Pi}_{\widetilde{\lambda}\mathbf{Q}}(\bar{z}, z'), \tag{77b}$$

where we use Eq. (50) and introduce the irreducible exciton self-energy

$$\widetilde{\Pi}_{\widetilde{\lambda}\mathbf{Q}}(z, \bar{z}) \equiv \sum_{\lambda'}\sum_{\mathbf{Q}'\alpha} |\widetilde{\mathcal{G}}_{\alpha-\mathbf{Q}'}^{\widetilde{\lambda}\lambda'}(\mathbf{Q})|^2\, N_{\lambda'\mathbf{Q}-\mathbf{Q}'}^{\mathrm{inc}}(z, \bar{z})\, D_{\alpha\mathbf{Q}'}(z, \bar{z}). \tag{78}$$

The irreducible exciton-phonon coupling in $\widetilde{\Pi}$ generalizes the one in Eq. (52) to finite momentum transfer:

$$\widetilde{\mathcal{G}}_{\alpha-\mathbf{Q}'}^{\widetilde{\lambda}\lambda'}(\mathbf{Q}) \equiv \sum_{c_1 c_2 v_1 \mathbf{k}_1} \widetilde{A}_{c_1 v_1 \mathbf{k}_1}^{\widetilde{\lambda}\mathbf{Q}*} g_{\alpha-\mathbf{Q}'}^{s, c_1 c_2}(\mathbf{k}_1 + \mathbf{Q}) A_{c_2 v_1 \mathbf{k}_1}^{\lambda'\mathbf{Q}-\mathbf{Q}'} - \sum_{c_1 v_1 v_2 \mathbf{k}_1} \widetilde{A}_{c_1 v_1 \mathbf{k}_1}^{\widetilde{\lambda}\mathbf{Q}*} g_{\alpha-\mathbf{Q}'}^{s, v_2 v_1}(\mathbf{k}_1 + \mathbf{Q}') A_{c_1 v_2 \mathbf{k}_1 + \mathbf{Q}'}^{\lambda'\mathbf{Q}-\mathbf{Q}'}. \tag{79}$$

The Matsubara component of the irreducible exciton self-energy agrees with the result in Ref. [73] if we replace $N^{\mathrm{inc}}$ with $N^{\mathrm{HSEX}}$ and $\widetilde{\mathcal{G}}$ with $\mathcal{G}$. As we see below, having a self-energy which is a functional of $N^{\mathrm{inc}}$ is essential to derive a Boltzmann-like equation.

To extract the equation of motion for the number of irreducible excitons we subtract Eq. (77b) from Eq. (77a) and then set $z = t_-$ and $z' = t_+$. Extending the MGKBA in Eq. (74) to the propagator $N^{\mathrm{inc}}$, i.e.,

$$N_{\lambda\mathbf{Q}}^{\mathrm{inc}, \lessgtr}(t, t') = \left[\theta(t - t')\, N_{\lambda\mathbf{Q}}^{\mathrm{inc}, \lessgtr}(t, t) + \theta(t' - t) N_{\lambda\mathbf{Q}}^{\mathrm{inc}, \lessgtr}(t', t')\right] e^{-iE_{\lambda\mathbf{Q}}(t - t')}, \tag{80}$$

and implementing the Markov approximation we obtain

$$\frac{d}{dt}\widetilde{N}_{\widetilde{\lambda}\mathbf{Q}}(t) = -\widetilde{\Gamma}_{\widetilde{\lambda}\mathbf{Q}}^{\mathrm{out}}(t)\widetilde{N}_{\widetilde{\lambda}\mathbf{Q}}(t) + \widetilde{\Gamma}_{\widetilde{\lambda}\mathbf{Q}}^{\mathrm{in}}(t)\left(1 + \widetilde{N}_{\widetilde{\lambda}\mathbf{Q}}(t)\right), \tag{81}$$

where the irreducible excitonic rates are given by

$$\widetilde{\Gamma}_{\widetilde{\lambda}\mathbf{Q}}^{\mathrm{out}} = 2\pi \sum_{\lambda'\alpha\mathbf{Q}'} \frac{\left|\widetilde{\mathcal{G}}_{\alpha-\mathbf{Q}'}^{\widetilde{\lambda}\lambda'}(\mathbf{Q})\right|^2}{2\omega_{\alpha\mathbf{Q}'}}\left(1 + N_{\lambda'\mathbf{Q}-\mathbf{Q}'}^{\mathrm{inc}}\right)$$
$$\times \left[\delta\left(E_{\lambda'\mathbf{Q}-\mathbf{Q}'} - \widetilde{E}_{\widetilde{\lambda}\mathbf{Q}} + \omega_{\alpha\mathbf{Q}'}\right)(1 + f_{\alpha\mathbf{Q}'}^{\mathrm{ph}}) + \delta\left(E_{\lambda'\mathbf{Q}-\mathbf{Q}'} - \widetilde{E}_{\widetilde{\lambda}\mathbf{Q}} - \omega_{\alpha\mathbf{Q}'}\right)f_{\alpha-\mathbf{Q}'}^{\mathrm{ph}}\right], \tag{82a}$$

$$\widetilde{\Gamma}_{\widetilde{\lambda}\mathbf{Q}}^{\mathrm{in}} = 2\pi \sum_{\lambda'}\sum_{\mathbf{Q}'\alpha} \frac{\left|\widetilde{\mathcal{G}}_{\alpha-\mathbf{Q}'}^{\widetilde{\lambda}\lambda'}(\mathbf{Q})\right|^2}{2\omega_{\alpha\mathbf{Q}'}} N_{\lambda'\mathbf{Q}-\mathbf{Q}'}^{\mathrm{inc}}$$
$$\times \left[\delta\left(E_{\lambda'\mathbf{Q}-\mathbf{Q}'} - \widetilde{E}_{\widetilde{\lambda}\mathbf{Q}} + \omega_{\alpha\mathbf{Q}'}\right)f_{\alpha\mathbf{Q}'}^{\mathrm{ph}} + \delta\left(E_{\lambda'\mathbf{Q}-\mathbf{Q}'} - \widetilde{E}_{\widetilde{\lambda}\mathbf{Q}} - \omega_{\alpha\mathbf{Q}'}\right)(1 + f_{\alpha-\mathbf{Q}'}^{\mathrm{ph}})\right]. \tag{82b}$$

Notice that this is *not* a Boltzmann equation for $\widetilde{N}$ since the rates depend on the occupations $N^{\mathrm{inc}}$.

### 5.3 Incoherent excitons

The number of incoherent excitons is given by the sum of $v$-reducible and $D$-reducible excitons, see Eq. (13) and (28):

$$N^{\text{inc}} = N^{(v)} + N^{(D)}. \tag{83}$$

According to the approximation in Eq. (72):

$$L^{(v)} = L^{\text{HSEX}} + iL^{\text{HSEX}}K^{\text{c}}[L]L. \tag{84}$$

The equation of motion for $N^{(v)}(t)$ can be derived from Eq. (84), by following the same logic as for $\widetilde{N}$. Considering only the diagonal elements of $N^{\text{inc}}$ in the excitonic basis and taking into account Eq. (41) we find

$$\frac{d}{dt}N^{(v)}_{\lambda\mathbf{Q}}(t) = -\Gamma^{\text{out}}_{\lambda\mathbf{Q}}(t)N^{\text{inc}}_{\lambda\mathbf{Q}}(t) + \Gamma^{\text{in}}_{\lambda\mathbf{Q}}(t)\big(1 + N^{\text{inc}}_{\lambda\mathbf{Q}}(t)\big), \tag{85}$$

where the excitonic rates

$$\Gamma^{\text{out}}_{\lambda\mathbf{Q}} = 2\pi \sum_{\lambda'\alpha\mathbf{Q}'} \frac{\big|\mathcal{G}^{\lambda\lambda'}_{\alpha-\mathbf{Q}'}(\mathbf{Q})\big|^2}{2\omega_{\alpha\mathbf{Q}'}}\big(1 + N^{\text{inc}}_{\lambda'\mathbf{Q}-\mathbf{Q}'}\big)$$
$$\times \Big[\delta\big(E_{\lambda'\mathbf{Q}-\mathbf{Q}'} - E_{\lambda\mathbf{Q}} + \omega_{\alpha\mathbf{Q}'}\big)(1 + f^{\text{ph}}_{\alpha\mathbf{Q}'}) + \delta\big(E_{\lambda'\mathbf{Q}-\mathbf{Q}'} - E_{\lambda\mathbf{Q}} - \omega_{\alpha\mathbf{Q}'}\big)f^{\text{ph}}_{\alpha-\mathbf{Q}'}\Big], \tag{86a}$$

$$\Gamma^{\text{in}}_{\lambda\mathbf{Q}} = 2\pi \sum_{\lambda'}\sum_{\mathbf{Q}'\alpha} \frac{\big|\mathcal{G}^{\lambda\lambda'}_{\alpha-\mathbf{Q}'}(\mathbf{Q})\big|^2}{2\omega_{\alpha\mathbf{Q}'}}N^{\text{inc}}_{\lambda'\mathbf{Q}-\mathbf{Q}'}$$
$$\times \Big[\delta\big(E_{\lambda'\mathbf{Q}-\mathbf{Q}'} - E_{\lambda\mathbf{Q}} + \omega_{\alpha\mathbf{Q}'}\big)f^{\text{ph}}_{\alpha\mathbf{Q}'} + \delta\big(E_{\lambda'\mathbf{Q}-\mathbf{Q}'} - E_{\lambda\mathbf{Q}} - \omega_{\alpha\mathbf{Q}'}\big)(1 + f^{\text{ph}}_{\alpha-\mathbf{Q}'})\Big], \tag{86b}$$

have the same mathematical form as the irreducible excitonic rates in Eq. (82), the difference being that the irreducible exciton energies and wavefunctions are replaced by the exciton ones. Notice the emergence of the original exciton-phonon coupling defined in Eq. (44).

### 5.4 Coherent excitons

The improved approximation to $\widetilde{L}$ leads to an improved equation of motion for the exciton polarization as well. In fact, the collision integral $S_\lambda$ must now be evaluated with $\widetilde{N}$, whose lesser component is, in general, nonzero. Using the MGKBA in Eq. (74) it is straightforward to derive

$$\frac{d}{dt}\rho_\lambda(t) = -iE_{\lambda\mathbf{0}}\rho_\lambda(t) - i\Omega_\lambda(t) - \frac{1}{2}\sum_{\widetilde{\lambda}'\mathbf{Q}}\Big[\widetilde{\Gamma}^{\text{pol}}_{\lambda\widetilde{\lambda}'\mathbf{Q}} + \widetilde{\Gamma}_{\lambda\widetilde{\lambda}'\mathbf{Q}}\widetilde{N}_{\widetilde{\lambda}'\mathbf{Q}}(t)\Big]\rho_\lambda(t), \tag{87}$$

which should be compared with Eq. (53).

## 6 Excitonic Bloch equations

In Table 1, we summarize the main results from Section 5, highlighting the types of excitons on which the various rates depend. We refer to the equations in Table 1 as the *excitonic Bloch equations* (XBE). Choosing the time origin earlier than the switch-on time of the external driving fields (hence $\Omega_\lambda(t < 0) = 0$), the XBE must be solved with initial conditions

| Definition | Equations |
|---|---|
| Exciton polarization | $\frac{d}{dt}\rho_\lambda = -iE_{\lambda\mathbf{0}}\rho_\lambda - i\Omega_\lambda - \frac{1}{2}\sum_{\widetilde{\lambda}'\mathbf{Q}}\left[\widetilde{\Gamma}^{\text{pol}}_{\lambda\widetilde{\lambda}'\mathbf{Q}} + \widetilde{\Gamma}_{\lambda\widetilde{\lambda}'\mathbf{Q}}\widetilde{N}_{\widetilde{\lambda}'\mathbf{Q}}\right]\rho_\lambda$ |
| Irreducible excitons | $\frac{d}{dt}\widetilde{N}_{\widetilde{\lambda}\mathbf{Q}} = -\widetilde{\Gamma}^{\text{out}}_{\widetilde{\lambda}\mathbf{Q}}[N^{\text{inc}}]\widetilde{N}_{\widetilde{\lambda}\mathbf{Q}} + \widetilde{\Gamma}^{\text{in}}_{\widetilde{\lambda}\mathbf{Q}}[N^{\text{inc}}]\left(1 + \widetilde{N}_{\widetilde{\lambda}\mathbf{Q}}\right)$ |
| Incoherent excitons | $\frac{d}{dt}N^{\text{inc}}_{\lambda\mathbf{Q}} = -\Gamma^{\text{out}}_{\lambda\mathbf{Q}}[N^{\text{inc}}]N^{\text{inc}}_{\lambda\mathbf{Q}} + \Gamma^{\text{in}}_{\lambda\mathbf{Q}}[N^{\text{inc}}]\left(1 + N^{\text{inc}}_{\lambda\mathbf{Q}}\right) + \sum_{\lambda'\widetilde{\lambda}}|S^{\mathbf{Q}}_{\lambda\widetilde{\lambda}}|^2\left[\widetilde{\Gamma}^{\text{pol}}_{\lambda'\widetilde{\lambda}\mathbf{Q}} + \widetilde{\Gamma}_{\lambda'\widetilde{\lambda}\mathbf{Q}}\widetilde{N}_{\widetilde{\lambda}\mathbf{Q}}\right]|\rho_{\lambda'}|^2$ |

Table 1: Excitonic Bloch equations.

$\rho_\lambda(0) = \widetilde{N}_{\widetilde{\lambda}\mathbf{Q}}(0) = N^{\text{inc}}_{\lambda\mathbf{Q}}(0) = 0$. The first line contains the equation of motion of the exciton polarization, see Eq. (87). The quantity

$$\gamma_\lambda \equiv \sum_{\widetilde{\lambda}'\mathbf{Q}}\left[\widetilde{\Gamma}^{\text{pol}}_{\lambda\widetilde{\lambda}'\mathbf{Q}} + \widetilde{\Gamma}_{\lambda\widetilde{\lambda}'\mathbf{Q}}\widetilde{N}_{\widetilde{\lambda}'\mathbf{Q}}\right] \tag{88}$$

gives the exciton linewidth of a photoabsorption spectrum, and agrees with Ref. [50] for low excitation densities (i.e., $\widetilde{N} = 0$). An increasing number of irreducible excitons accelerate the transition toward the incoherent regime, in agreement with the fact that the polarization lifetime decreases with the excitation density [38,77]. The equation of motion for the number of irreducible excitons, see Eq. (81), is shown in the second line. The third line contains the equation of motion for the number of incoherent excitons, obtained by adding Eqs. (75) and (85). Coherent excitons are first converted into $D$-reducible excitons $N^{(D)}$, see Eq. (75), which then diffuse and becomes incoherent excitons. Aside from the overscreening issue, the equation of motion for $N^{\text{inc}}$ agrees with findings from the cluster expansion [12,53,54] in the limit of infinitesimal excitation densities. This limit corresponds to set $\widetilde{N} = 0$ and to retain only terms linear in $N^{\text{inc}}$. In the incoherent regime, i.e., $\rho_\lambda = 0$, the equation of motion for $N^{\text{inc}}$ also agrees with findings from Ref. [78]. We observe, however, that setting $\rho_\lambda = 0$ from the outset yields a homogeneous equation, meaning that the initial value of the incoherent exciton population must be determined by other means. The many-body diagrammatic treatment of the ab initio Hamiltonian not only provides a first-principles justification of the work in Ref. [78], but also extends it to the coherent regime. This extension allows for monitoring the exciton dynamics from the moment the optical field drives the system out of equilibrium.

The XBE have the merits of being overscreening free and applicable to nonequilibrium systems beyond the linear response regime. Furthermore, they preserve Eq. (66), according to which the total number of excitons remains constant when the driving field is off.

# 7   Conclusions

Starting from the ab initio Hamiltonian for electrons and phonons [60] and using the many-body diagrammatic Green's function theory [64] we have derived a first principles scheme for material specific predictions of nonequilibrium excitons. The XBE are a system of nonlinear differential equations for the coupled dynamics of coherent excitons, irreducible excitons and incoherent excitons. They encompass the initial transient regime, driven by external optical fields, as well as the evolution from the coherent to the incoherent regime, governed by

exciton-phonon scatterings. Importantly, the XBE are *free* of overscreening issues. It is worth remarking that the first-principles formulation presented in this work involves only screened *e-ph* couplings. In Appendix B we outline an alternative first-principles formulation that eliminates the need for introducing irreducible excitons but involves the bare *e-ph* coupling. In the ab initio theory of electrons and phonons the bare $g$ appears in the exact formula of the phonon self-energy [65, 79, 80] as well as in the coupling to coherent phonons [60, 67, 81]. However, in the present context the use of a bare $g$ complicates the formulation since both intraband and interband *e-ph* coupling must be accounted for. In fact, the dressing of $g^{cc'}$ and $g^{vv'}$ is mainly due to $g^{cv}$ and $g^{vc}$.

The XBE form a minimal set of equations for describing exciton formation, diffusion and thermalization for not too high excitation densities. They can be improved along several directions that we wish to discuss here. First, the lifting of the Markov approximation through the introduction of higher order correlators. This idea has been already implemented in nonequilibrium systems of electrons [32, 76, 82–84] and bosons [38, 85–87] for several many-body approximations, and it is also possible to implement it in this context. Second, the extension to nonequilibrium phonons. In this work, we have assumed that phonons remain in thermal equilibrium. However, at sufficiently high excitation densities, this assumption becomes unrealistic. The equation of motion for the phonon occupations can be derived following the approach outlined in Ref. [67]. However, the phononic self-energy must be refined to account for excitonic effects in the polarization. Third, at large enough excitation density the Coulomb mediated exciton-exciton scattering can no longer be ignored. This interaction has been investigated in Refs. [88–90], and it gives rise to additional rates in the equations of motion. Fourth, excitons can strongly couple to coherent optical phonons. This interaction is governed by the bare *e-ph* coupling [60, 81] and is responsible for a time-dependent shift of the exciton energies. Given the long timescale of optical nuclear displacements, we expect that an adiabatic approximation of the XBE, i.e., $E_{\lambda \mathbf{Q}} \to E_{\lambda \mathbf{Q}}(t)$, is reasonably accurate. Fifth, the inclusion of exciton recombination through the quantized treatment of photons. This aspect has been already covered in, e.g., Ref. [53], and does not present criticalities as the ab initio Hamiltonian for electrons and photons has long been well established.

We hope that our contribution can clarify the pitfalls inherent in model Hamiltonians, serve as a solid ground for the development of a rigorous theory of nonequilibrium excitons, phonons and photons, and inspire parameter-free numerical schemes for real-time simulations of excitonic materials.

## Acknowledgements

**Funding information**    This work has been supported by MIUR PRIN (Grant No. 2022WZ8LME), INFN through the TIME2QUEST project and Tor Vergata University through Project TESLA.

## A    On the polarization rates

For the evaluation of Eq. (43) we assume that the Green's function $G_{\mu\mu'\mathbf{k}}$ with band indices $\mu, \mu'$ either both conduction or both valence is diagonal. Let us analyze $\left[\Sigma^{\mathrm{FM}}(z,\bar{z})G(\bar{z},z^+)\right]_{cv\mathbf{k}}$. Focusing solely on the dependence on the electronic band indices, this term contains either $g^{s,cc_1}G_{c_1c_1}(z,\bar{z})g^{s,c_1c_2}G_{c_2v}(\bar{z},z^+)$ or $g^{s,cc_1}G_{c_1v_1}(z,\bar{z})g^{s,v_1v}G_{vv}(\bar{z},z^+)$. The second diagram in Fig. 1

contributes as $\left[\Sigma^{(1)}(z,\bar{z})G(\bar{z},z^+)\right]_{cv\mathbf{k}}$. We have the following possible structures

$$g^{s,cc_1}G_{c_1c_1}(z,\bar{z})G_{c_2c_2}(\bar{z}',\bar{z}'')g^{s,c_2c_3}G_{c_3v_1}(\bar{z}'',\bar{z}')G_{vv}(\bar{z},z), \tag{89a}$$

$$g^{s,cc_1}G_{c_1v_1}(z,\bar{z})G_{v_2c_2}(\bar{z}',\bar{z}'')g^{s,c_2c_3}G_{c_3c_3}(\bar{z}'',\bar{z}')G_{c_4v}(\bar{z},z), \tag{89b}$$

$$g^{s,cc_1}G_{c_1c_1}(z,\bar{z})G_{c_2v_1}(\bar{z}',\bar{z}'')g^{s,v_1v_2}G_{v_2v_2}(\bar{z}'',\bar{z}')G_{vv}(\bar{z},z), \tag{89c}$$

$$g^{s,cc_1}G_{c_1v_1}(z,\bar{z})G_{v_2v_2}(\bar{z}',\bar{z}'')g^{s,v_2v_3}G_{v_3c_2}(\bar{z}'',\bar{z}')G_{c_3v}(\bar{z},z). \tag{89d}$$

We see that the first and third combinations are linear in the off-diagonal $G$ whereas the second and third combinations are cubic, and can therefore be ignored. We finally consider $\left[\Sigma^{(2)}(z,\bar{z})G(\bar{z},z^+)\right]_{cv\mathbf{k}}$. We have the following possible structure

$$G_{c_1c_1}(\bar{z}',\bar{z}'')g^{c_1c_2}G_{c_2v_2}(\bar{z}'',\bar{z}')G_{v_3c_3}(z,\bar{z})g^{c_3c_4}G_{c_4v}(\bar{z},z), \tag{90a}$$

$$G_{c_1c_1}(\bar{z}',\bar{z}'')g^{c_1c_2}G_{c_2v_2}(\bar{z}'',\bar{z}')G_{v_3v_3}(z,\bar{z})g^{v_3v}G_{vv}(\bar{z},z), \tag{90b}$$

$$G_{c_1v_1}(\bar{z}',\bar{z}'')g^{v_1v_2}G_{v_2v_2}(\bar{z}'',\bar{z}')G_{v_3c_3}(z,\bar{z})g^{c_3c_4}G_{c_4v}(\bar{z},z), \tag{90c}$$

$$G_{c_1v_1}(\bar{z}',\bar{z}'')g^{v_1v_2}G_{v_2v_2}(\bar{z}'',\bar{z}')G_{v_3v_3}(z,\bar{z})g^{v_3v}G_{vv}(\bar{z},z). \tag{90d}$$

The first and third combinations are cubic in the off-diagonal $G$, and we discard them. The second and fourth combinations contain $G_{v_3v_3}(z,\bar{z})G_{vv}(\bar{z},z)$ which scales linearly with the excitation density, and we therefore discard these combinations as well. We can use the ansatz in Eqs. (46) to express the off-diagonal elements of $\widetilde{L}$ in terms of the irreducible exciton propagator:

$$\widetilde{L}^{\mathbf{Q}}_{cv\mathbf{k},c_1c_2\mathbf{k}_1}(z,\bar{z};z,\bar{z}) = \sum_{v_1}\widetilde{N}^{\mathbf{Q}}_{cv\mathbf{k},c_1v_1\mathbf{k}_1}(z,\bar{z})\rho_{c_2v_1\mathbf{k}_1}(\bar{t}), \tag{91a}$$

$$\widetilde{L}^{\mathbf{Q}}_{cv\mathbf{k},v_2v_1\mathbf{k}_1}(z,\bar{z};z,\bar{z}) = -\sum_{c_1}\widetilde{N}^{\mathbf{Q}}_{cv\mathbf{k},c_1v_1\mathbf{k}_1}(z,\bar{z})\rho_{c_1v_2\mathbf{k}_1+\mathbf{Q}}(\bar{t}). \tag{91b}$$

After some algebra we find

$$\left[\Sigma(z,\bar{z})G(\bar{z},z^+)\right]_{cv\mathbf{k}} = i\sum_{c_1c_2c_3v_1,\alpha\mathbf{Q}}g^{s,cc_1}_{\alpha-\mathbf{Q}}(\mathbf{k})\widetilde{N}^{-\mathbf{Q}}_{c_1v\mathbf{k},c_2v_1\mathbf{k}_1}(z,\bar{z})\rho_{c_3v_1\mathbf{k}_1}(\bar{t})g^{s,c_2c_3}_{\alpha\mathbf{Q}}(\mathbf{k}_1-\mathbf{Q})D_{\alpha\mathbf{Q}}(z,\bar{z})$$
$$-i\sum_{c_1c_2v_1v_2,\alpha\mathbf{Q}}g^{s,cc_1}_{\alpha-\mathbf{Q}}(\mathbf{k})\widetilde{N}^{-\mathbf{Q}}_{c_1v\mathbf{k},c_2v_2\mathbf{k}_1}(z,\bar{z})\rho_{c_2v_1\mathbf{k}_1-\mathbf{Q}}(\bar{t})g^{s,v_1v_2}_{\alpha\mathbf{Q}}(\mathbf{k}_1-\mathbf{Q})D_{\alpha\mathbf{Q}}(z,\bar{z}). \tag{92}$$

We now contract the left hand side with the exciton wave functions, and expand $\widetilde{N}$ in the irreducible excitonic basis and $\rho$ in the excitonic basis

$$\sum_{cv\mathbf{k}}A^{\lambda\mathbf{0}*}_{cv\mathbf{k}}\left[\Sigma(z,\bar{z})G(\bar{z},z^+)\right]_{cv\mathbf{k}} = i\sum_{\widetilde{\lambda}_1\lambda_2\alpha\mathbf{Q}}\left(\sum_{cc_1v\mathbf{k}}A^{\lambda\mathbf{0}*}_{cv\mathbf{k}}g^{s,cc_1}_{\alpha-\mathbf{Q}}(\mathbf{k})\widetilde{A}^{\widetilde{\lambda}_1-\mathbf{Q}}_{c_1v\mathbf{k}}\right)\widetilde{N}_{\widetilde{\lambda}_1-\mathbf{Q}}(z,\bar{z})\rho_{\lambda_2}(\bar{t})D_{\alpha\mathbf{Q}}(z,\bar{z})$$

$$\times\left(\sum_{c_2c_3v_2\mathbf{k}_1}\widetilde{A}^{\widetilde{\lambda}_1-\mathbf{Q}*}_{c_2v_1\mathbf{k}_1}g^{s,c_2c_3}_{\alpha\mathbf{Q}}(\mathbf{k}_1-\mathbf{Q})A^{\lambda_2\mathbf{0}}_{c_3v_1\mathbf{k}_1}\right)$$

$$-i\sum_{\widetilde{\lambda}_1\lambda_2\alpha\mathbf{Q}}\left(\sum_{cc_1v\mathbf{k}}A^{\lambda\mathbf{0}*}_{cv\mathbf{k}}g^{s,cc_1}_{\alpha-\mathbf{Q}}(\mathbf{k})\widetilde{A}^{\widetilde{\lambda}_1-\mathbf{Q}}_{c_1v\mathbf{k}}\right)\widetilde{N}_{\widetilde{\lambda}_1-\mathbf{Q}}(z,\bar{z})\rho_{\lambda_2}(\bar{t})D_{\alpha\mathbf{Q}}(z,\bar{z})$$

$$\times\left(\sum_{c_2v_2v_3\mathbf{k}_1}\widetilde{A}^{\widetilde{\lambda}_1-\mathbf{Q}*}_{c_2v_2\mathbf{k}_1}g^{s,v_3v_2}_{\alpha\mathbf{Q}}(\mathbf{k}_1-\mathbf{Q})A^{\lambda_2\mathbf{0}}_{c_2v_3\mathbf{k}_1-\mathbf{Q}}\right). \tag{93}$$

It is useful to define the following quantities

$$\sum_{c_1 c_2 v_1 \mathbf{k}_1} \widetilde{A}_{c_1 v_1 \mathbf{k}_1}^{\widetilde{\lambda} \mathbf{Q}*} g_{\alpha - \mathbf{Q}'}^{s, c_1 c_2}(\mathbf{k}_1 + \mathbf{Q}) A_{c_2 v_1 \mathbf{k}_1}^{\lambda' \mathbf{Q} - \mathbf{Q}'} \equiv \widetilde{g}_{\alpha - \mathbf{Q}'}^{(c)\widetilde{\lambda}\lambda'}(\mathbf{Q}), \tag{94a}$$

$$\sum_{c_1 c_2 v_1 \mathbf{k}_1} A_{c_1 v_1 \mathbf{k}_1}^{\lambda \mathbf{Q} - \mathbf{Q}'*} g_{\alpha \mathbf{Q}'}^{s, c_1 c_2}(\mathbf{k}_1 + \mathbf{Q} - \mathbf{Q}') \widetilde{A}_{c_2 v_1 \mathbf{k}_1}^{\widetilde{\lambda}' \mathbf{Q}} = \widetilde{g}_{\alpha - \mathbf{Q}'}^{(c)\widetilde{\lambda}'\lambda*}(\mathbf{Q}), \tag{94b}$$

$$\sum_{c_1 v_1 v_2 \mathbf{k}_1} \widetilde{A}_{c_1 v_1 \mathbf{k}_1}^{\widetilde{\lambda} \mathbf{Q}*} g_{\alpha - \mathbf{Q}'}^{s, v_2 v_1}(\mathbf{k}_1 + \mathbf{Q}') A_{c_1 v_2 \mathbf{k}_1 + \mathbf{Q}'}^{\lambda' \mathbf{Q} - \mathbf{Q}'} \equiv \widetilde{g}_{\alpha - \mathbf{Q}'}^{(v)\widetilde{\lambda}\lambda'}(\mathbf{Q}), \tag{94c}$$

$$\sum_{c_1 v_1 v_2 \mathbf{k}_1} A_{c_1 v_1 \mathbf{k}_1 + \mathbf{Q}'}^{\lambda \mathbf{Q} - \mathbf{Q}'*} g_{\alpha \mathbf{Q}'}^{s, v_2 v_1}(\mathbf{k}_1) \widetilde{A}_{c_1 v_2 \mathbf{k}_1}^{\widetilde{\lambda}' \mathbf{Q}} = \widetilde{g}_{\alpha - \mathbf{Q}'}^{(v)\widetilde{\lambda}'\lambda*}(\mathbf{Q}), \tag{94d}$$

where we use the property $g_{\alpha - \mathbf{Q}}^{\mu \nu}(\mathbf{k}) = g_{\alpha \mathbf{Q}}^{\nu \mu *}(\mathbf{k} - \mathbf{Q})$, see Eq. (20). Then we can rewrite Eq. (93) as

$$\sum_{cv\mathbf{k}} A_{cv\mathbf{k}}^{\lambda 0*} \big[ \Sigma(z, \bar{z}) G(\bar{z}, z^+) \big]_{cv\mathbf{k}} = i \sum_{\widetilde{\lambda}_1 \lambda_2} \sum_{\alpha \mathbf{Q}} \widetilde{g}_{\alpha \mathbf{Q}}^{(c)\widetilde{\lambda}_1 \lambda *}(-\mathbf{Q}) \widetilde{N}_{\widetilde{\lambda}_1 - \mathbf{Q}}(z, \bar{z})$$
$$\times \big[ \widetilde{g}_{\alpha \mathbf{Q}}^{(c)\widetilde{\lambda}_1 \lambda_2}(-\mathbf{Q}) - \widetilde{g}_{\alpha \mathbf{Q}}^{(v)\widetilde{\lambda}_1 \lambda_2}(-\mathbf{Q}) \big] \rho_{\lambda_2}(\bar{t}) D_{\alpha \mathbf{Q}}(z, \bar{z}). \tag{95}$$

Proceeding along the same lines we can show that

$$\sum_{cv\mathbf{k}} A_{cv\mathbf{k}}^{\lambda 0*} \big[ G(z, \bar{z}) \Sigma(\bar{z}, z^+) \big]_{cv\mathbf{k}} = i \sum_{\widetilde{\lambda}_1 \lambda_2} \sum_{\alpha \mathbf{Q}} \widetilde{g}_{\alpha \mathbf{Q}}^{(v)\widetilde{\lambda}_1 \lambda *}(-\mathbf{Q}) \widetilde{N}_{\widetilde{\lambda}_1 - \mathbf{Q}}(z, \bar{z})$$
$$\times \big[ \widetilde{g}_{\alpha \mathbf{Q}}^{(c)\widetilde{\lambda}_1 \lambda_2}(-\mathbf{Q}) - \widetilde{g}_{\alpha \mathbf{Q}}^{(v)\widetilde{\lambda}_1 \lambda_2}(-\mathbf{Q}) \big] \rho_{\lambda_2}(\bar{t}) D_{\alpha \mathbf{Q}}(z, \bar{z}). \tag{96}$$

The collision integral $S_\lambda$ is the difference between Eqs. (95) and (96). Using the definition of the irreducible exciton-phonon coupling in Eq. (52) we conclude that

$$S_\lambda(t) = i \sum_{\widetilde{\lambda}_1 \lambda_2} \sum_{\alpha \mathbf{Q}} \int d\bar{z} \, \widetilde{\mathcal{G}}_\alpha^{\widetilde{\lambda}_1 \lambda *}(\mathbf{Q}) \widetilde{N}_{\widetilde{\lambda}_1 \mathbf{Q}}(z, \bar{z}) \widetilde{\mathcal{G}}_\alpha^{\widetilde{\lambda}_1 \lambda_2}(\mathbf{Q}) \rho_{\lambda_2}(\bar{t}) D_{\alpha - \mathbf{Q}}(z, \bar{z}), \tag{97}$$

where we rename $\mathbf{Q} \to -\mathbf{Q}$.

We now observe that for any Keldysh function $k(z, z')$

$$\int d\bar{z} \, k(z, \bar{z}) = \int d\bar{t} \, k^{\mathrm{R}}(t, \bar{t}). \tag{98}$$

Taking into account that for the product of functions [64]

$$[\widetilde{N} D]^{\mathrm{R}}(t, \bar{t}) = \widetilde{N}^{\mathrm{R}}(t, \bar{t}) D^>(t, \bar{t}) + \widetilde{N}^<(t, \bar{t}) D^{\mathrm{R}}(t, \bar{t}), \tag{99}$$

we can rewrite Eq. (103) as

$$S_\lambda(t) = i \sum_{\widetilde{\lambda}_1 \lambda_2} \sum_{\alpha \mathbf{Q}} \widetilde{\mathcal{G}}_\alpha^{\widetilde{\lambda}_1 \lambda *}(\mathbf{Q}) \widetilde{\mathcal{G}}_\alpha^{\widetilde{\lambda}_1 \lambda_2}(\mathbf{Q})$$
$$\times \int^t d\bar{t} \big[ \widetilde{N}_{\widetilde{\lambda}_1 \mathbf{Q}}^{\mathrm{R}}(t, \bar{t}) D_{\alpha - \mathbf{Q}}^<(t, \bar{t}) + \widetilde{N}_{\widetilde{\lambda}_1 \mathbf{Q}}^<(t, \bar{t}) D_{\alpha - \mathbf{Q}}^{\mathrm{R}}(t, \bar{t}) \big] \rho_{\lambda_2}(\bar{t}). \tag{100}$$

To evaluate this quantity we use the MGKBA in Eq. (74) for $\widetilde{N}$, the dressed $D$ of the Born-Oppenheimer approximation [67]

$$D_{\alpha \mathbf{Q}}^{\lessgtr}(t, t') = \frac{\theta(t - t')}{2i\omega_{\alpha \mathbf{Q}}} \big[ n_{\alpha \mathbf{Q}}^{\lessgtr}(t) e^{-i\omega_{\alpha \mathbf{Q}}(t - t')} + n_{\alpha - \mathbf{Q}}^{\gtrless}(t) e^{i\omega_{\alpha \mathbf{Q}}(t - t')} \big]$$
$$+ \frac{\theta(t' - t)}{2i\omega_{\alpha \mathbf{Q}}} \big[ n_{\alpha \mathbf{Q}}^{\lessgtr}(t') e^{-i\omega_{\alpha \mathbf{Q}}(t - t')} + n_{\alpha - \mathbf{Q}}^{\gtrless}(t') e^{i\omega_{\alpha \mathbf{Q}}(t - t')} \big], \tag{101}$$

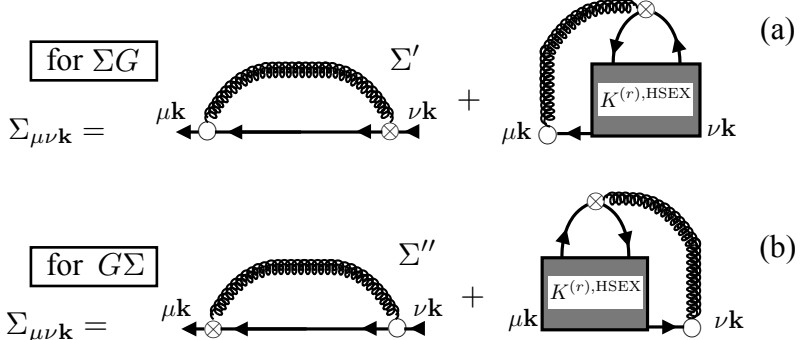

Figure 4: Overscreening-free approximation of the phononic self-energy leading to a formulation in terms of bare *e-ph* couplings.

where $n_{\alpha\mathbf{Q}}^{<} = f_{\alpha\mathbf{Q}}^{\mathrm{ph}}$ and $n_{\mathbf{Q}\alpha}^{>}(t) = f_{\alpha\mathbf{Q}}^{\mathrm{ph}} + 1$, and approximate

$$\rho_\lambda(t') = e^{-iE_{\lambda 0}(t'-t)}\rho_\lambda(t), \tag{102}$$

see Eq. (53). A technical remark. The self-energy in Fig. 1 *is not positive definite* in the sense of Ref. [91], and it gives rise to complex rates for nonvanishing phononic coherences $\Theta_{\alpha\mathbf{Q}} = \Theta_{\alpha-\mathbf{Q}}(t) = \langle \hat{b}_{\alpha\mathbf{Q}}\hat{b}_{\alpha-\mathbf{Q}}\rangle$. We therefore discard phononic coherence in our treatment. Taking the Markovian limit we find

$$S_\lambda(t) = \frac{1}{2}\sum_{\widetilde{\lambda}'\lambda''\mathbf{Q}}\left[\widetilde{\Gamma}_{\lambda\widetilde{\lambda}'\lambda''\mathbf{Q}}^{\mathrm{pol}}(t) + \widetilde{\Gamma}_{\lambda\widetilde{\lambda}'\lambda''\mathbf{Q}}(t)\widetilde{N}_{\widetilde{\lambda}'\mathbf{Q}}(t)\right]\rho_{\lambda''}(t), \tag{103}$$

where

$$\widetilde{\Gamma}_{\lambda\widetilde{\lambda}'\lambda''\mathbf{Q}}^{\mathrm{pol}} = 2\pi\sum_\alpha \frac{\widetilde{\mathcal{G}}_\alpha^{\widetilde{\lambda}'\lambda*}(\mathbf{Q})\widetilde{\mathcal{G}}_\alpha^{\widetilde{\lambda}'\lambda''*}(\mathbf{Q})}{2\omega_{\alpha\mathbf{Q}}}\left[\delta(\widetilde{E}_{\widetilde{\lambda}'\mathbf{Q}} - E_{\lambda''\mathbf{0}} + \omega_{\alpha\mathbf{Q}})n_{\alpha-\mathbf{Q}}^{>} + \delta(\widetilde{E}_{\widetilde{\lambda}'\mathbf{Q}} - E_{\lambda''\mathbf{0}} - \omega_{\alpha\mathbf{Q}})n_{\alpha\mathbf{Q}}^{<}\right], \tag{104a}$$

$$\widetilde{\Gamma}_{\lambda\widetilde{\lambda}'\lambda''\mathbf{Q}} = 2\pi\sum_\alpha \frac{\widetilde{\mathcal{G}}_\alpha^{\widetilde{\lambda}'\lambda*}(\mathbf{Q})\widetilde{\mathcal{G}}_\alpha^{\widetilde{\lambda}'\lambda''*}(\mathbf{Q})}{2\omega_{\alpha\mathbf{Q}}}\left[\delta(\widetilde{E}_{\widetilde{\lambda}'\mathbf{Q}} - E_{\lambda''\mathbf{0}} + \omega_{\alpha\mathbf{Q}}) - \delta(\widetilde{E}_{\widetilde{\lambda}'\mathbf{Q}} - E_{\lambda''\mathbf{0}} - \omega_{\alpha\mathbf{Q}})\right]. \tag{104b}$$

The result reduces to Eqs. (54) and (76) when retaining only the contribution $\lambda'' = \lambda$.

# B   Irreducible excitons or bare *e-ph* coupling?

The challenge in formulating the theory using excitons instead of irreducible excitons lies in the appearance of the product of bare and screened *e-ph* couplings in $\Sigma G = g^s D g L^{\mathrm{HSEX}}$, see discussion below Eq. (45). Since dynamical effects in the dressing of $g$ are typically disregarded, the Markov limit would result in non-positive rates. One possible strategy to formulate the theory exclusively in terms of excitons is to work with bare *e-ph* couplings.

In Appendix A we have shown that for small excitation densities only $\Sigma^{\mathrm{FM}} + \Sigma^{(1)}$ contributes to $[\Sigma G]_{cv\mathbf{k}}$, and only $\Sigma^{\mathrm{FM}} + \Sigma^{(2)}$ contributes to $[G\Sigma]_{cv\mathbf{k}}$. Let us slightly modify the self-energy. For $[\Sigma G]_{cv\mathbf{k}}$ we use $\Sigma' = igDGg^s$, instead of $\Sigma^{\mathrm{FM}} = ig^s DGg^s$, and evaluate $\Sigma^{(1)}$ with a bare $g$ on the left (keeping the screened $g^s$ at the top), see Fig. 4(a). Then

$$\Sigma G = igD\ell g^s + igD(\widetilde{L}^{\mathrm{SEX}} - \ell)g^s = igD\widetilde{L}^{\mathrm{SEX}}(1 - iv L^{\mathrm{HSEX}})g = igDL^{\mathrm{HSEX}}g, \tag{105}$$

where in the last equality we use Eq. (27). Similarly, for $[G\Sigma]_{cv\mathbf{k}}$ we use $\Sigma'' = ig^s GDg$ , instead of $\Sigma^{\text{FM}} = ig^s DGg^s$, and evaluate $\Sigma^{(2)}$ with a bare $g$ on the right (keeping the screened $g^s$ at the top). We then find

$$G\Sigma = ig^s G\ell Dg + ig^s(\widetilde{L}^{\text{SEX}} - \ell)Dg = g(1 - L^{\text{HSEX}}\nu)\widetilde{L}^{\text{SEX}}g = igDL^{\text{HSEX}}g. \tag{106}$$

Following the same steps as in Section 3 the equation of motion for the exciton polarization becomes

$$\frac{d}{dt}\rho_\lambda = -iE_{\lambda\mathbf{0}}\rho_\lambda - i\Omega_\lambda - \frac{1}{2}\sum_{\lambda'\mathbf{Q}}\widetilde{\Gamma}^{\text{pol}}_{\lambda\lambda'\mathbf{Q}}(t)\rho_\lambda(t), \tag{107}$$

where the polarization rates are expressed in terms of excitonic energies and wavefunctions

$$\widetilde{\Gamma}^{\text{pol}}_{\lambda\lambda'\mathbf{Q}} = 2\pi\sum_\alpha \frac{|\mathcal{G}^{b,\lambda'\lambda}_\alpha(\mathbf{Q})|^2}{2\omega_{\alpha\mathbf{Q}}}\Big[\delta(E_{\lambda'\mathbf{Q}} - E_{\lambda\mathbf{0}} + \omega_{\alpha\mathbf{Q}})(1 + f^{\text{ph}}_{\alpha-\mathbf{Q}}) + \delta(E_{\lambda'\mathbf{Q}} - E_{\lambda\mathbf{0}} - \omega_{\alpha\mathbf{Q}})f^{\text{ph}}_{\alpha\mathbf{Q}}\Big], \tag{108}$$

and $\mathcal{G}^b$ is the *bare* exciton-phonon couplings

$$\mathcal{G}^{b,\lambda\lambda'}_\alpha(\mathbf{Q}) \equiv \sum_{c_1 c_2 v_1 \mathbf{k}_1} A^{\lambda\mathbf{Q}*}_{c_1 v_1 \mathbf{k}_1} g^{c_1 c_2}_{\alpha-\mathbf{Q}}(\mathbf{k}_1 + \mathbf{Q})A^{\lambda'\mathbf{0}}_{c_2 v_1 \mathbf{k}_1} - \sum_{c_1 v_1 v_2 \mathbf{k}_1} A^{\lambda\mathbf{Q}*}_{c_1 v_1 \mathbf{k}_1} g^{v_2 v_1}_{\alpha-\mathbf{Q}}(\mathbf{k}_1 + \mathbf{Q})A^{\lambda'\mathbf{0}}_{c_1 v_2 \mathbf{k}_1+\mathbf{Q}}. \tag{109}$$

The equation of motion Eq. (64) for the incoherent exciton numbers can also be reformulated in terms of only excitons and bare *e-ph* couplings. According to Eq. (30) we have $L = L^{(\nu)} - iL^{(\nu)}gDgL^{(\nu)}$. Using this expression and following the same steps as in Section 4 we find

$$\frac{d}{dt}N^{\text{inc}}_{\lambda\mathbf{Q}} = \sum_{\lambda'}\Gamma^{\text{coh}}_{\lambda'\lambda\mathbf{Q}}(t)|\rho_{\lambda'}(t)|^2. \tag{110}$$

It is straightforward to verify that also in in this alternative formulation the total number of excitons satisfies Eq. (66).

The above formulation overlooks an important detail. For a semiconductor in equilibrium at low temperature it is reasonable to set to zero all matrix elements of $L$ except for $L_{cv,c'v'}$ and $L_{vc,v'c'}$. From Eq. (29) we have (omitting the dependence on momenta and times)

$$g^{s,cc'} = g^{cc'} - i\nu_{c\bar{a}ac'}\sum_{\alpha\bar{\alpha}\beta\bar{\beta}}\nu_{c\bar{a}ac'}L_{\alpha\bar{\alpha},\beta\bar{\beta}}g^{\beta\bar{\beta}}. \tag{111}$$

We see that the dressing of $g^{cc}$ is due to the interband bare *e-ph* couplings $g^{cv}$ and $g^{vc}$. The same holds true for the dressing of $g^{vv'}$. Since the interband couplings are not negligible, they must be included. This complicates the whole theory as the equations of motion can no longer be closed solely on exciton numbers.

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
