# Peer review of "Excitonic Bloch equations from first principles"

_SciPost Physics_

## Round 1 · Referee Report · Anonymous (Referee 1) · 2024-9-20

Report

In this manuscript, starting from basic equations of many body Green's function theory, the authors provide a derivation of the excitonic Bloch equations beyond that based on model Hamiltonians. In this way they provide a theoretical tool for the description of the exciton dynamics that can be interfaced with modern ab-initio calculations of electronic structure and lattice vibrations. In addition their derivation suggests that the exciton dynamics governed by model Hamiltonians is affected by an overscreening of the electron-phonon interaction. The problem of the overscreening seems to be removed in their formulation.

In my opinion the manuscript sounds physically clear, well explained and I am in favor of its publication. However I found that there are some issues in the formulation of the theory and some questions the author should answer before the paper is published.

\begin{enumerate}
\item I am not really convinced that the definition of the exciton-phonon coupling provided in previous works such as in Ref. [73,75] of the present manuscript is affected by overscreening. From basic equations of many body Green's function theory, the Bethe-Salpeter equation (BSE) in presence of electron-phonon interaction can be obtained once an approximation for the electron self-energy in terms of the electron-phonon coupling is given. In particular, the functional derivative of the electron self-energy respect to the electron Green's function gives the kernel of the BSE from which we can extract an effective exciton self-energy that provides also the definition of the exciton-phonon coupling as discussed in Ref. [50] of the present manuscript. Still in that work the authors have shown that a suitable approximation for the electron self-energy is that reported in Fig. 1 (b) of the present manuscript. Indeed in this expression, as mentioned by the authors, the electron-phonon coupling is not overscreened in contrast with the expression reported in Fig.1 (a) where the overscreening is clear. Thus, I totally agree with authors that the correct expression of the electron self-energy is that reported in Fig. 1 (b). At this point, it is important to note that the structure of the exciton self-energy (and hence the exciton-phonon coupling) depends from the way in which the functional derivative of the electron self-energy is performed. In Ref. [50] the functional derivative is done neglecting the explicit dependence from the electron Green's function of $K^{(r),SEX}$ and the electron-phonon matrix elements $g^s$. This leads to an expression of the exciton self-energy in terms of the proper part of the tow-particle correlation function $\tilde{L}^{SEX}$ instead of the full $L^{HSEX}$ as in Ref. [73,75]. This is the origin of the asymmetric structure of the exciton-phonon coupling $\tilde{\mathcal{G}}$ in Eq. (52). However, in principle there is no reason to neglect the functional derivative of $g^s$ which depends from the electron Green's function through the screening. This leads to the appearance of additional diagrams in the exciton self-energy. In particular, I suspect that, when the screening is evaluated using $L^{HSEX}$ (i.e. consistently with our treatment of vertex corrections), the inclusion of the terms arising from $\frac{\delta g^s}{\delta G}$ would lead to the natural appearance of $L^{HSEX}$ in the exciton self-energy and thus to a symmetric exciton-phonon coupling $\mathcal{G}$ as in Eq. (44).

In the present manuscript, an expression of the exciton-phonon matrix element formally equivalent to that of Ref. [50] has been obtained evaluating the collision integral in Eq. (43) using the ansatz in Eq. (46) and taking only terms linear in the off-diagonal part of $G$. However in doing this, the authors consider only off-diagonal $G$ appearing in the external lines. But what about the linear terms where the off-diagonal $G$ appear inside $g^s$? I suspect that neglecting these terms is in some way equivalent to neglect $\frac{\delta g^s}{\delta G}$ in the derivation of Ref. [50].

I suggest to think about this point before resubmitting the manuscript.

\item The eigen-value problem in Eq. (2) describes excitons in absence of population. Thus it can be used only for optical absorption. In the present case the authors are treating excitons out-of-equilibrium. What about the Pauli blocking factors. Is there a reason to neglect them in Eq. (2)? Is this related to some assumption concerning the laser pulse (frequency of the laser pulse close to the exciton energy, small population etc)?

The author should clarify this point in the manuscript.

\item The Bloch equations obtained in this work should describe how photoexcited coherent excitons are converted into incoherent excitons and how the latter propagate and eventually thermalize. Thus when the quasi-equilibrium is reached, these equations predict a photoexcited system consisting of a thermalized incoherent exciton gas. However, in general we expect a configuration in which incoherent excitons coexist with an electron-hole plasma [see for example: phys. stat. sol. (b) 131, 151 (1985)]. What about the electron-hole plasma? What is the regime in which the theory developed by the authors is applicable?

I think the author should discuss these aspects in the article.

\item In Eq. (2) I do not see any index associated to the spin degrees of freedom. Thus, I suppose that spin variables have been summed up in some way. This requires the introduction of two decoupled BSEs. One for the singlet channel and one for the triplet channel [see: RIVISTA DEL NUOVO CIMENTO
VOL. 11, N. 12]. If this is the case Eq. (2) refer to the singlet channel [see Phys. Rev. B 62, 4927 (2000)]. As a consequence, the matrix elements of the bar Coulomb potential in Eq. (3) should be multiplied for a factor 2.

\item In the manuscript the author call $v$ "direct electron-hole interaction" and $W$ "exchange electron-hole interaction". In the literature, on the contrary, direct and exchange are usually used to indicate $W$ and $v$, respectively.

\end{enumerate}

Attachment

Recommendation

Ask for minor revision

  • validity: -
  • significance: -
  • originality: -
  • clarity: -
  • formatting: -
  • grammar: -

Author:  Gianluca Stefanucci  on 2024-10-21  [id 4883]

(in reply to Report 1 on 2024-09-20)

We provide a reply to the Referee's comments in the attached file

Attachment:

reply.pdf

---

## Round 1 · Referee Report · Anonymous (Referee 2) · 2024-10-9

Report

This work presents a consistent treatment for the dynamics of bound electron-hole pairs out of equilibrium, in terms of a set of “excitonic” Bloch equations driven by the coupling with lattice vibrations. These equations account for both the coherent and incoherent dynamics of the excitons, as well as the conversion between excitonic polarizations and populations. In addition, the authors work out the microscopic forms of the exciton-phonon couplings discussing the complex role played by electronic screening.

I think this work represents a relevant contribution to both the theories of nonequilibrium carrier dynamics and of exciton-phonon interaction, presenting very timely results leading to a step forward in the theoretical framing of these processes. Thus, in my opinion the manuscript should appear on SciPost Physics.

That being said, I do have several questions and comments. I do not think the work is always easy to follow, and sometimes it is not especially clear what has been deduced via derivations and what constitutes a somewhat arbitrary approximation. I would ask the authors to please clarify these points and, whenever they feel the discussion can be beneficial to the manuscript, to consider amending the text.

  1. The BSE has a spin structure exhibiting different channels: excitons (singlet and triplet) plus magnons. In the case of collinear spin polarization, singlet and triplet excitons are mixed. In the case of spin-orbit interaction, excitons and magnons are mixed. [See Bechstedt, Many-Body Approach to Electronic Excitations, Chapter 18.2] The theory presented here is most likely applicable to the exciton channel (singlet+triplet) for all kinds of spin polarizations and including spin-orbit coupling, while neglecting the magnon channel. The authors could explicitly remark on this.

  2. The authors could remark on the connection between the exciton polarization in Eq. (15), which is derived from the disconnected part of the two-particles Green’s function, and photoabsorption spectra. This could be useful since in most textbook derivations of the BSE, which are in the context of linear response and Hedin’s equations, photoabsorption spectra are obtained starting from the Dyson’s equation for L instead of GG. [See e.g. Strinati, Rivista del Nuovo Cimento, 11, 12 (1988)]

  3. The theory is formulated in the Tamm-Dancoff approximation, in which the Coulomb couplings between resonant and antiresonant valence-conduction transitions are neglected in the BSE kernel. It is presently unclear how much these terms are important out of equilibrium (even for a semiconductor), but I suspect that if they are included in the phonon-assisted kernel, they will also lead to the explicit appearance of the interband valence-conduction (screened) electron-phonon couplings inside additional exciton-phonon contributions. Do the authors have any comments on this?

  4. The authors distinguish between the coherent exciton regime, where the exciton-phonon coupling is “irreducible”, and the incoherent regime, where it is “reducible”. Excitonic polarizations are then converted to populations again via the irreducible exciton coupling with phonons. The overscreening discussion in the coherent case, along with the identification of the correct electron-phonon self-energy and irreducible ex-ph coupling, is equivalent to the findings in Ref. [50] starting from linear response. The discussion of the exciton-phonon BSE kernel and relative reducible ex-ph coupling for the incoherent case is equivalent to Refs. [75] and [Cudazzo, PRB 102, 045136 (2020)] starting from an extension of the static BSE kernel (here I propose that the latter reference may be cited in the text). However, while these cited papers assumed that either the irreducible or reducible couplings should be used in all cases, the present paper makes the case for both formulations being correct, albeit in different regimes. It could be useful to explicitly state this in the text in order to make the points of convergence and departure with previous works clearer. I do have some additional comments on this.

4.1 The authors emphasize that the coherent part of the theory could be expressed in terms of reducible excitons if bare el-ph couplings were used. In the text they write “$g_S \tilde{L} = g L^{(v)}$”, but the implication seems to be – by looking at Appendix B – that actually the $v$-reducible $L$ could be replaced with $L^{HSEX}$, i.e. only the first term in Eq. (67). Is this correct? This means that the exciton-phonon self-energy obtained with the irreducible ex-ph coupling and the irreducible $\tilde{L}$ should be equivalent to the one obtained with the bare ex-ph coupling and the reducible – actually HSEX – $L$. But $L^{HSEX}$ and $\tilde{L}$ have different poles. How can Eq. (88) still be valid (including in the low-density regime) in the $g$$ L^{HSEX}$ formulation? In particular, in Ref. [50] it is pointed out that the $g_S$ * $\tilde{L}$ treatment gives rise to intrinsic nonzero phonon-mediated linewidths also for the lowest-lying excitonic state. I don’t understand how a treatment with $g$$ L^{HSEX}$ could give the same.

4.2 About the incoherent excitons. In this case, the authors do not start from an electronic self-energy, but rather from the BSE for $L$ (the correlated part of the two-particles Green’s function) with a general interaction kernel, which is then approximated to HSEX for the Coulomb part, and to first order in $D$ (the phonon Green’s function) for the phonon part. Are these choices compelled by consistency with the previous coherent treatment (such as: approximating $\tilde{L}$ as $L^{SEX}$ requires $L$ to become $L^{HSEX}$) and/or compliance with state-of-the art simulations (since BSE calculations are usually done in the HSEX approximation)? Is it not possible to obtain an electronic self-energy whose functional derivative with respect to the Green’s function would yield this HSEX+ph kernel? And if not, can there be consistency issues (such as missed cancellations) between terms arising from the dressing of the electron Green’s functions (quasiparticle corrections) and those appearing in the incoherent exciton-phonon kernel?

4.3 On a related note: does Eq. (66) for the conservation of the total exciton number remain valid independently of the approximation chosen for $N^{inc}$ in Eq. (65), that is, the type of incoherent kernel employed?

  1. Does $N^{inc}(t)$ tend to relax to a Bose-Einstein distribution after long times? What about $\tilde{N}(t)$? In other words, can this theory provide some hints at the form of the “exciton” occupation function after they have relaxed to the bottom of their dispersion curves, before recombination?

Spotted typos: - Eq. (7): the last $z^+$ should be $z^{\prime +}$ - Before Eq. (50): $N^{HSEX}$ in the text should be $\tilde{N}^{SEX}$

Recommendation

Ask for minor revision

  • validity: top
  • significance: top
  • originality: high
  • clarity: good
  • formatting: perfect
  • grammar: excellent

Author:  Gianluca Stefanucci  on 2024-10-21  [id 4884]

(in reply to Report 2 on 2024-10-09)

We provide a reply to the Referee's comments in the attached file

Attachment:

reply_wrjjmUJ.pdf

---

## Editorial Decision

resubmitted